# Electron pairing in mirror modes: Surpassing the quasilinear limit

**Rudolf A. Treumann**[1,3] **and Wolfgang Baumjohann**[2]

[1]International Space Science Institute, Bern, Switzerland
[2]Space Research Institute, Austrian Academy of Sciences, Graz, Austria
[3]Geophysics Department, Ludwig-Maximilians-University Munich, Germany,
*Correspondence to*: Wolfgang.Baumjohann@oeaw.ac.at

**Abstract.** The mirror mode evolving in collisionless magnetised high-temperature thermally anisotropic plasmas is shown to develop an interesting macro-state. Starting as a classical zero frequency ion fluid instability it saturates quasi-linearly at very low magnetic level, while forming elongated magnetic bubbles which trap the electron component to perform an adiabatic bounce motion along the magnetic field. Further evolution of the mirror mode towards a stationary state is determined by the bouncing trapped electrons which interact with the thermal level of ion sound waves, generate attractive wake potentials which give rise to formation of electron pairs in the lowest-energy singlet state of two combined electrons. Pairing takes preferentially place near the bounce-mirror points where the pairs become spatially locked with all their energy in the gyration. The resulting large anisotropy of pairs enters the mirror growth rate in the quasi-linearly stable mirror mode. It breaks the quasilinear stability and causes further growth. Pressure balance is either restored by dissipation of the pairs and their anisotropy or inflow of plasma from the environment. In the first case new pairs will continuously form until equilibrium is reached. In the final state the fraction of pairs can be estimated. This process is open to experimental verification. To our knowledge it is the only process where in high temperature plasma pairing may occur and has an important observable macroscopic effect: breaking the quasilinear limit and, via pressure balance, generation of localised diamagnetism.

**Keywords.** Mirror modes, magnetosheath, solar wind, turbulence, plasma diamagnetism

## 1 Introduction

There seems to be nothing particularly interesting left about a very low frequency effect in high temperature magnetised plasma known as the mirror mode (see, e.g., Tsuru-

tani et al., 2011, for a more recent observational review). It was formally discovered some sixty years ago (Chandrasekhar, 1961; Hasegawa, 1975; Gary, 1993) as a theoretical complement to the zero-frequency hose instability, two purely growing linear instabilities in the presence of pressure anisotropies. The hose instability excites propagating Alfvén waves when the magnetically parallel temperature $T_\parallel > T_\perp$ exceeds the perpendicular temperature, the mirror mode grows under the opposite condition $T_\parallel < T_\perp$ that the perpendicular temperature is higher than the parallel by a certain amount, passing a threshold. The mirror mode generates magnetically elongated magnetic bottles in pressure balance, thereby providing the plasma a local texture. In the presence of weak plasma gradients, the mirror mode, formally, assumes a small but finite real frequency (Hasegawa, 1969). Various properties of the instability were added under different plasma conditions and in different wavenumber ranges like finite gyroradius effects (cf., e.g., Pokhotelov et al., 2005), dependencies on electron temperature and electron anisotropies (cf., e.g., Pokhotelov et al., 2003), as well as on plasma convection. The instability saturates quasilinearly at a rather low level by exhausting the bulk thermal anisotropy (cf., e.g., Treumann & Baumjohann, 1997; Noreen et al., 2017, for the quasilinear numerics). Finally, the trapped particle components give rise to the excitation of ion cyclotron and electron whistler waves, if only a thermal anisotropy of *resonant* particles evolves. Identification in real plasmas became possible when measuring the pressure balance between external magnetic field and increased internal plasma pressure. In observations the two pressures in ion-mirror modes are anti-correlated, a condition which generally serves as the key identifier of mirror modes.

The disturbing remaining barely understood point is how, in an ideally conducting plasma at high temperature, the magnetic field can become expelled to a high degree from the interior of the magnetic mirror bottle, an effect re-

sembling the Meissner effect in low-temperature superconductivity which, however, is forbidden in classical physics as it requires the presence of quantum correlations known to be restricted to very dense and low-temperature conditions only. In superconductivity, the Meissner effect arises from the electron-phonon interaction in the crystal lattice when spin-compensated electron-Cooper pairs (Cooper, 1956) form, occupy the same quantum state and, though remaining fermions, together with the interacting phonons assume bosonic properties.

These quasi-particles condensate in the lowest energy level above Fermi energy and evolve into a Landau-Fermi fluid (Landau, 1941; Ginzburg & Landau, 1950; Ginzburg, 1955), culminating in BCS theory (Bardeen et al., 1957). The condensate current becomes capable of generating up to $\sim 100\%$ bulk diamagnetism, rather different from pressure balance which in BCS is warranted by the stiffness of the crystal lattice. In mirror modes the quasi-diamagnetic effect amounts to roughly 50%, many times more than the $\sim$ few % magnetic amplitudes quasilinear theory predicts (cf., e.g., Noreen et al., 2017). For its explanation, no weak kinetic plasma turbulence theory (cf., e.g., Yoon, 2007) is in sight. This discrepancy suggests that in the evolution of mirror modes some fundamental effect is still missing.

Recently we exhumed kind of a parallelism between superconductivity and the growth of the ion mirror mode (Treumann & Baumjohann, 2018a). Here we demonstrate that the mirror mode can be understood as a combination of the classical plasma ion effect which generates magnetic bottles at the low quasilinear saturation level, while the main large mirror effect may possibly be caused by the trapped bouncing particle component in a similar though classical way as in the BCS theory.

In this case it are the mirroring particles (preferentially electrons, though possibly also ions), which interact with the available general thermal ion-sound wave population in the plasma, at either thermal or non-thermal level, to produce trapped pair singlets which are dynamically distributed over the volume of a mirror bottle. Together with the ion sound fluctuations, they form kind of quasiparticles. These become locked near their centre-of-mass mirror points and condensate at perpendicular energy close to the electron temperature. This leads to electron boundary currents causing some weak diamagnetism which, probably, is incapable of explaining the further growth of mirror modes.

However, evolving from the quasilinear stable state, the pairs generate a large perpendicular electron anisotropy that enters the mirror growth rate and breaks the quasilinear stability. Further mode growth is either pressure compensated by dissolution of pairs and plasma heating or by quasineutral plasma inflow from the environment along the magnetic field. This kind of pairing is proper to the mirror instability and may develop for electrons and, possibly, under modified conditions and effects, also for ions. For being specific, we concentrate on electrons only in the following.

The physical mechanism behind this effect is the interaction of mirror-mode-trapped electrons with the thermal background noise of the ion-acoustic wave spectrum excited in the mirror unstable plasma at low but sufficiently large amplitude. Since ion sound is a basic plasma eigenmode, balance between spontaneous emission and collisionless damping always leads to the presence of ion sound noise at measurable intensity, for instance in the magnetosheath (cf., e.g., Rodriguez & Gurnett, 1975; Treumann & Baumjohann, 2018b, for additional arguments concerning mirror mode observations). Hence, for the interaction discussed in this communication it is not necessary that the ion sound is excited by an instability. If the conditions for an ion-acoustic instability are satisfied, the pairing condition may become positively affected.

## 2 Electron trapping

Once the ion-mirror mode starts growing at the well known (cf., e.g., Noreen et al., 2017) ion-mirror growth rate

$$\frac{\gamma_m(\boldsymbol{k})}{\omega_{ci}} \approx \frac{k_\parallel \lambda_i}{1+A}\sqrt{\frac{\beta_\parallel}{\pi}}\left[A+\sqrt{\frac{T_{e\perp}}{T_{i\perp}}}A_e - \frac{k^2}{k_\perp^2 \beta_\perp}\right] \quad (1)$$

$$A \equiv \frac{T_{i\perp}}{T_{i\parallel}} - 1 \gtrsim 0, \qquad A_e \equiv \frac{T_{e\perp}}{T_{e\parallel}} - 1$$

with approximately vanishing real frequency $\omega_m \approx 0$, neglecting the effect of density gradients, which would cause a finite real part on the frequency, $\lambda_i = c/\omega_i$ ion inertial length, a magnetic bottle evolves in slightly oblique direction $k_\parallel \ll k_\perp$ with magnetic disturbances $|\delta B_\parallel| \gg |\delta B_\perp|$. This bottle is elongated or stretched along the ambient magnetic field $\boldsymbol{B}$ and has a narrow opening angle $\theta$ given by $\tan \theta = k_\parallel/k_\perp \ll 1$. Instability corresponds to a second order phase transition in plasma which happens whence the magnetic field locally drops below a critical threshold value

$$B < B_c \approx \sqrt{2\mu_0 N T_{i\perp} A}\,|\sin \theta| \quad (2)$$

where we neglected the electron contribution. Though substantial, this growth rate is just a fraction of the ion cyclotron frequency $\omega_{ci} = eB/m_i$ for $k_\parallel \lambda_i \ll 1$ and $B$ respectively $A$ near threshold, the usual case (Treumann & Baumjohann, 2018a). As noted above, the instability readily stabilises quasi-linearly (Noreen et al., 2017; Treumann & Baumjohann, 1997) at very low level $|\delta B|^2 \ll B^2$ via depleting the anisotropy $A$. One may note that a marginally stable case is obtained for

$$A\beta_\perp = k^2/k_\perp^2 \quad (3)$$

which defines a particular marginally stable angle of propagation.

More interesting is that, for an initially negligible electron contribution $A_e \ll A$, the electron anisotropy at quasilinear

stability may become important in the growth rate for the mirror instability to surpass the stable quasilinear state, in which case the ion mirror instability may further grow.

It is thus of particular interest, whether a substantial electron anisotropy $A_e$ can self-consistently be generated in the mirror mode as this could cause growth of the ion mirror mode to large amplitudes.

Below we propose such a mechanism. We may, however, note in passing that this same mechanism can also, by itself, drive an electron mirror mode after quasilinear stability of the ion mode has been reached. The electron mirror mode is a short-scale magnetic fluctuation on the ion mode background with fast growth rate

$$\frac{\gamma_e}{\omega_{ce}} \approx k_\parallel \lambda_e \sqrt{\frac{\beta_{e\parallel}}{\pi}} \left[ A_e - \frac{k^2}{k_\perp^2 \beta_{e\perp}} \right] \qquad (4)$$

a fraction of the electron cyclotron frequency $\omega_{ce}$. The marginal stability condition again requires vanishing of the bracket for one particular angle of propagation. The electron mode also stabilises quasi-linearly, causing short wavelength depletions of the magnetic field on the course of the above ion mirror mode, as have been observed previously (cf., e.g., Treumann & Baumjohann, 2018b).

In the following we do not further investigate the electron mirror mode, even though it will also be affected by the proposed electron pairing which might be responsible for its surprisingly large amplitudes it achieves. Our focus here is on the main ion mode.

### 2.1 Electron dynamics, energy limit, trapped density fraction

The conventional ion mirror mode provides a quasi-stationary magnetic bottle (see, e.g., Constantinescu, 2002, for a analytical geometric model) structure, which necessarily traps electrons of sufficiently small magnetic moment $\mu_e$. Because the mirror mode frequency practically vanishes and the mirror mode grows slowly compared to the electron dynamics, electrons react adiabatically to the presence of the mirror instability. They conserve their magnetic moment $\mu_e = \mathcal{E}_{e\perp}/B = $ const when moving along the magnetic field $B(s)$. Trapping occurs between the two mirror magnetic fields $B_{\pm m} = B(\pm s_m)$, with $s$ coordinate along the magnetic field. The trapped-electron perpendicular kinetic energy $\mathcal{E}_{e\perp}(s) = \mu_e B(s) \equiv V(s)$ plays the role of a retarding potential

$$\mathcal{E}_{e\parallel}(s) = \mathcal{E}_e - V(s) \qquad (5)$$

At the mirror points the parallel energy of trapped electrons vanishes, and $\mathcal{E}_{e\perp}(\pm s_m) = \mathcal{E}_e$. Thus trapping occurs for all $\mu_e \leq \mu_m \equiv \mathcal{E}_e/B_{\pm m}$, a well-known fact. Though it does not bunch them, mirroring keeps these electrons together by confining them to the volume of the bottle, inside which they perform the oscillatory bounce motion between mirror points $\pm s_m$. The parallel electron equation of motion is

$$\frac{dv_\parallel(s,t)}{dt} = -\frac{\mu_e}{m_e} \nabla_\parallel B(s), \qquad \nabla_\parallel = \frac{\partial}{\partial s} \qquad (6)$$

For symmetric bottles and motion around and not too far away from the minimum $B(s_0) = \min\{B(s)\} \equiv B_0$ of the magnetic field we have

$$B(s) \approx B_0 + \tfrac{1}{2} B_0''(s - s_0)^2, \qquad B_0'' = \frac{\partial^2 B}{\partial s^2}\bigg|_{s_0} \qquad (7)$$

which immediately gives the bounce frequency

$$\omega_b = \sqrt{\frac{\mu_e}{m_e} B_0''} \ll \omega_{ce} \qquad (8)$$

a frequency much less than the electron cyclotron frequency $\omega_{ce} = eB/m_e$.

We shall show below that this kind of trapping, in the case of the mirror mode, becomes advantageous for electron pairing, an effect otherwise observed only under solid state conditions in superconducting metals.

In order to get an idea on the trapping energy condition, we consider the mirror point $s = \pm s_m$. Here all the energy is in the perpendicular direction, i.e. the local gyro motion of electrons. Hence de-trapping of electrons occurs, once their gyroradius exceeds the opening radius of the bottle neck $r_{ce,s} \gtrsim R_s = \mathcal{L}_\parallel \tan\theta$, where $\mathcal{L}_\parallel \sim 2\pi/k_\parallel$ is the half length of the ion mirror bottle. This yields immediately that electrons remain trapped as long as their energy satisfies the condition

$$\mathcal{E}_e \lesssim \frac{1}{4\pi^2} \frac{T_e}{k_\perp^2 r_{ce}^2} \equiv \mathcal{E}_{trap}, \qquad r_{ce} = v_e/\omega_{ce,s} \qquad (9)$$

with $r_{ce}$ is the electron gyro-radius and $k_\perp$ the perpendicular wave number of the ion-mirror mode. All electrons of such energy remain trapped in the magnetic mirror bottle. Larger energy electrons escape from the bottle along the magnetic field. (We do not discuss the subtle problem that quasi-neutrality requires them to be replaced by low-perpendicular-speed electron inflow along the magnetic field which, however, has an effect on the additionally required pressure balance.)

### 2.2 Fractional trapping condition

The fractional number density of maxwellian mirror trapped electrons is

$$
\begin{aligned}
\frac{N_{trap}}{N_0} &= C \int_0^{\mathcal{E}_{trap}} d\epsilon_e \, \epsilon_e^{\frac{1}{2}} \exp\left(-\frac{\epsilon_e}{T_e}\right) \\
&= \Gamma^{-1}\left(\tfrac{3}{2}\right) \gamma\left(\tfrac{3}{2}, \frac{\mathcal{E}_{trap}}{T_e}\right) \\
&= \frac{1}{4} \mathrm{erf}\left(\sqrt{\frac{\mathcal{E}_{trap}}{T_e}}\right) - \sqrt{\frac{\mathcal{E}_{trap}}{T_e}} \exp\left(-\frac{\mathcal{E}_{trap}}{T_e}\right)
\end{aligned}
\qquad (10)
$$

where $C$ is a normalisation constant, and $\gamma(a,b)$ is the incomplete Gamma function.

Constancy of the trapped electron magnetic moment (cf., e.g., Baumjohann & Treumann, 1996, for a textbook presentation), yields the parallel energy

$$\mathcal{E}_\parallel(s) = \mathcal{E}_e \left( 1 - \frac{B(s)}{B(s_m)} \right) \qquad (11)$$

which defines the angle between velocity and magnetic field for the trapped electrons

$$\theta(s) = \cos^{-1} \sqrt{1 - \frac{B(s)}{B(s_m)}} \qquad (12)$$

These or the equivalent expressions we will need below.

## 3  Single electron wake potential

In preparing for the investigation of ion-mirror-mode trapped electrons, we consider the interaction of an electron with the bath of ion sound waves. This is most easily done in the naked test particle picture, assuming that we grab one of the electrons and ask for its reaction to the presence of the dielectric in which it moves. This approach requires subsequent integration over the electron energy distribution.

The electron is a point charge $-e$ with velocity $\boldsymbol{v}$ that is located at its instantaneous position $\boldsymbol{x}' = \boldsymbol{x} - \boldsymbol{v}t$ in the observers frame $(\boldsymbol{x},t)$. This is represented by the point charge density function $N(\boldsymbol{x},t) = -e\delta(\boldsymbol{x} - \boldsymbol{v}t)$. We assume that the electron is non-relativistic which for trapped electrons under the conditions in the magnetosheath (Lucek et al., 2005) or the solar wind is good enough. The relative dielectric constant of the plasma it experiences is $\epsilon(\omega,\boldsymbol{k})$ where $\omega,\boldsymbol{k}$ are frequency and wavenumber of the plasma wave which changes the dielectric properties. In general, we have a whole spectrum of waves which is taken care of below by integrating over the entire spectrum. The naked charged electron polarises the plasma. The total electric potential the moving non-relativistic charge at location $\boldsymbol{x}'$ causes at location $\boldsymbol{x}$ is obtained from Poisson's equation with above charge density and has the form

$$\Phi(\boldsymbol{x},t) = -\frac{e}{(2\pi)^3 \epsilon_0} \int d\omega d\boldsymbol{k} \frac{\delta(\omega - \boldsymbol{k}\cdot\boldsymbol{v})}{k^2 \epsilon(\boldsymbol{k},\omega)} e^{i\boldsymbol{k}\cdot(\boldsymbol{x}-\boldsymbol{v}t)} \qquad (13)$$

This can easily be shown (originally given by Neufeld & Ritchie, 1955; Krall & Trivelpiece, 1973, for a textbook description) by Fourier-transformation. One may note that this expression also holds for ions. In this representation the action of the $\delta$-function on the exponential has been taken care of. Integration is over wave numbers and frequencies, the wave spectrum responsible for the dielectric properties experienced by the test electron. Integration with respect to frequency $\omega$ implies $\omega \to \boldsymbol{k}\cdot\boldsymbol{v}$ also in the dielectric response

function $\epsilon(\omega,\boldsymbol{k})$, which we shift until having discussed the latter.

In solid state physics it is assumed that the oscillations of the ion lattice generate a thermal spectrum of phonons. In plasmas these waves are not restricted to the Brillouin zones but are freely propagating waves either forming a thermal background noise or, for $T_e \gg T_i$ providing a broad spectrum of unstably excited ion sound for which the plasma response function accounts. In high temperature thermal equilibrium it is the jitter motion of electrons which leads to spontaneous emission of sound and modifies the dielectric properties of the plasma. The general electrostatic response function reads

$$\epsilon(\omega,\boldsymbol{k}) = 1 + \frac{1}{k^2\lambda_e^2} + \chi_e(\omega,\boldsymbol{k}) + \chi_i(\omega,\boldsymbol{k}) \qquad (14)$$

with $\chi_{e,i}(\omega,\boldsymbol{k})$ the electron and ion susceptibilities. Under nonlinear conditions the susceptibilities depend, in addition, on the wave amplitude. This electrostatic dispersion relation contains both the effects of electrons and ions.

One may wonder why for wavelengths usually much longer than the Debye length $\lambda_e \ll \lambda$ the second term in this expression is not neglected. The reason that it must be retained here, is that the uncompensated charge of the test particle when immersed into the plasma excites short wavelengths waves on the Debye scale in order to screen the charge. Therefore, independent of the wavelength of plasma waves, the test particle dielectric response must include the Debye term.

The dielectric response function of the thermal spectrum of ion-sound waves at frequencies far below the electron plasma frequency $\omega \ll \omega_e$ is

$$\epsilon(\omega,\boldsymbol{k}) = 1 + \frac{1}{k^2\lambda_e^2} - \left(\frac{\omega_i}{\omega}\right)^2 \qquad (15)$$

where $\omega_i$ is the ion plasma frequency, $\lambda_e \approx v_e/\omega_e$ the Debye screening distance, and the frequency of ion sound waves $\omega_{\boldsymbol{k}}$ is obtained putting the real part of this expression to zero, which as usually yields

$$\left(\frac{\omega_{\boldsymbol{k}}}{\omega_i}\right)^2 = \frac{k^2\lambda_e^2}{1+k^2\lambda_e^2} \qquad \text{or} \qquad \omega_{\boldsymbol{k}}^2 = \frac{c_s^2 k^2}{1+k^2\lambda_e^2} < \omega_i^2 \qquad (16)$$

Here $c_s^2 = \omega_i^2 \lambda_e^2 \approx (m_e/m_i)v_e^2 \approx 2T_e/m_i$ is the ion-sound speed square. It is simple matter to show that the inverse response function becomes

$$\frac{1}{\epsilon(\omega,\boldsymbol{k})} = \frac{k^2\lambda_e^2}{1+k^2\lambda_e^2}\left(1 + \frac{\omega_{\boldsymbol{k}}^2}{\omega^2 - \omega_{\boldsymbol{k}}^2}\right) \qquad (17)$$

Actually, this is also the general inverse form of any dielectric response, if only $\omega_{\boldsymbol{k}}$ is understood as the solution of the general response function

$$\epsilon(\omega,\boldsymbol{k}) = 0 \qquad (18)$$

for electrostatic waves, and

$$\epsilon(\omega, \boldsymbol{k}) - k^2 c^2/\omega^2 = 0 \tag{19}$$

for very-low frequency electromagnetic waves like magneto-sonic or (kinetic) Alfvén waves. In the latter case one has (cf., e.g., Treumann & Baumjohann, 1997), including kinetic effects,

$$\epsilon_A(\omega, \boldsymbol{k}) = 1 + \frac{1}{k^2 \lambda_e^2} + \frac{c^2}{V_A^2}\left[1 + \left(\boldsymbol{k}\cdot\boldsymbol{r}_{ci}\right)^2\left(\frac{3}{4} + \frac{T_e}{T_i}\right)\right]^{-1} \tag{20}$$

with $\boldsymbol{r}_{ci} = \boldsymbol{v}_{i\perp}/\omega_{ci}$ the vectorial ion gyro-radius. The relevant wave frequency is $\omega_{\boldsymbol{k}A}^2 \approx k^2 V_A^2$ for the ordinary Alfvén wave, with $V_A \ll c$ the Alfvén speed (if wanted including the bracketed modification factor).

The kinetic Alfvén wave propagates fast along the magnetic field almost at Alfvén speed, and roughly ten times slower perpendicular to it. The weak wave electric potential resulting from its kinetic nature is along the magnetic field and is being believed to be responsible for electron acceleration. Any attractive pairing effect in resonance with those fast parallel electrons will be in this direction as well, a very interesting fact in itself which we do not investigate here, leaving it for a separate investigation. It may be applicable to particles in the auroral magnetosphere with its magnetic trapping configuration.

Inserting Eq. (17) into the above electrostatic potential of the test electron

$$\begin{aligned}\Phi(\boldsymbol{x}, t) &= -\frac{e\lambda_e^2}{(2\pi)^2\epsilon_0}\int\frac{d\omega\,k_\perp dk_\perp dk_\parallel}{1 + k^2\lambda_e^2}\times\\ &\times\left(1 + \frac{\omega_{\boldsymbol{k}}^2}{\omega^2 - \omega_{\boldsymbol{k}}^2}\right)\delta\big(\omega - \boldsymbol{k}\cdot\boldsymbol{v}\big)e^{i\boldsymbol{k}\cdot(\boldsymbol{x}-\boldsymbol{v}t)}\end{aligned} \tag{21}$$

shows that $\Phi$ consists quite generally of two contributions, the screened Coulomb potential of the test electron, and another wave induced term which multiplies the screened potential by the frequency dependent term in the last expression. (We note again that a similar form trivially holds as well for ions.) This form demonstrates the well known self-screening Debye effect of the naked point charge, which leads to the first term in the above expression and causes the Debye-Yukawa potential to exponentially compensate for the electron charge field in a spherical region of radius $\lambda_e$. We are not interested here in the deformation of the Debye sphere introduced by the electron motion as this is a higher order effect.

The zero order effect of the test electron contained in the wave-independent term, the proper self-screening is, in the wave-dependent term, multiplied by the wave-induced factor. For frequencies $\omega^2 = \boldsymbol{k}\cdot\boldsymbol{v}^2 > \omega_{\boldsymbol{k}}^2$ higher than ion sound, this factor is positive adding to the screening but changes sign for frequencies $\omega^2 = \boldsymbol{k}\cdot\boldsymbol{v}^2 < \omega_{\boldsymbol{k}}^2$, thereby indicating the possibility of over-screening at wavelengths larger than the Debye radius $\lambda_e$ (cf. Treumann & Baumjohann, 2014, their Fig. 1).

Under certain conditions it may come into play outside the Debye radius where the charge-electric field is practically already compensated, and the long range wave electric field adds up over some distance, may dominate and cause a spatially restricted deficiency of repulsion. In this case the potential may even turn negative, eliminates the repulsive nature of the electron locally and becomes attractive for electrons. This was first shown (Neufeld & Ritchie, 1955) for high frequency Langmuir waves even before the discovery of Cooper pairs in superconductivity and solid state physics. In a bath of Langmuir waves this attraction turned out to be unimportant however, while in an isotropic non-magnetic plasma it survives for low-frequency ion sound, first suggested by Nambu & Akama (1985). With $\theta_k$ the angle between electron speed and wavenumber, it happens at resonant electron speeds

$$v^2\cos^2\theta_k \lesssim \omega_{\boldsymbol{k}}^2/k^2 \tag{22}$$

requiring the parallel electron speed to be less than the wave phase velocity. The above expression depends on angle $\theta_k$ between velocity $\boldsymbol{v}$ and wavenumber $\boldsymbol{k}$, which in our case will turn out to be of crucial importance.

For completeness we note that in magnetised plasma the ion acoustic wave is azimuthally symmetric with respect to the magnetic field $\boldsymbol{B}$. However, its frequency depends itself on the angle of propagation between $\boldsymbol{k} = (k_\perp, k_\parallel)$ and $\boldsymbol{B}$ according to (Baumjohann & Treumann, 1996)

$$\omega_{\boldsymbol{k}}^2 = \frac{c_s^2\Lambda_0(\eta_i)k_\parallel^2}{\Lambda_0(\eta_e) + k^2\lambda_e^2} \tag{23}$$

with $\Lambda_0(\eta_j) = I_0(\eta_j)\exp(-\eta_j)$, $\eta_j = \frac{1}{2}k_\perp^2 r_{cj}^2$, and the index $j = e, i$ on the gyroradius is for electrons and ions. $I_0(\eta_j)$ is the Bessel function of imaginary argument. $r_{cj} = v_{\perp,j}/\omega_{cj}$ is the gyroradius, and $\omega_{cj}$ is the cyclotron frequency. One has that, moreover, $k_\perp\lambda_e \ll k_\perp r_{ci} \ll 1$ and $k_\parallel/k_\perp < 1$. Long-wavelength ion sound in magnetised plasma thus propagates essentially along the magnetic field, a well known fact which in observations, for instance in the magnetosheath (Rodriguez & Gurnett, 1975), manifests itself as a complete drop out of the electrostatic low frequency thermal ion noise spectrum when the antenna points strictly perpendicular to the ambient magnetic field (cf., e.g., Treumann & Baumjohann, 2018b, for an example and discussion).

The interaction between electrons and ion sound waves thus opens up the option that electrons in a Debye-screened potential may, under certain conditions, experience an attractive potential which compensates and overcomes the Coulomb repulsion between two negatively charged electrons, resembling a the famous effect of Cooper pairing in solid state physics though here in the realm of classical physics. The paired electrons and the propagating ion sound wave form a quasiparticle in both these cases.

It is important to insist that this attraction is not due to trapping of the electron by a large amplitude wave in the

wave potential trough; on the contrary, it is an electron-induced change in the dielectric properties of the wave-carrying plasma causing the electron to evolve an attractive electrostatic wake potential which it carries along when moving across the plasma. We have previously shown (Treumann & Baumjohann, 2014) that this can happen also with other waves than ion-sound. Below we demonstrate that it becomes crucial in the evolution of mirror modes to which plasma wave trapping does not contribute in no sense.

Since the waves are propagating along the magnetic field and the bounce motion of the electrons is as well along the magnetic field, the coordinate $s$ of interest is parallel to the magnetic field $\hat{s}\|B$, and the gyration of the electrons decouples from the interaction. In this case we have for the wave number $k = (k_\|, k_\perp)$ and velocity

$$k_\| \equiv \boldsymbol{k} \cdot \hat{\boldsymbol{s}}, \qquad v_\|(s) \equiv \boldsymbol{v} \cdot \hat{\boldsymbol{s}} = v \cos\theta(s)$$

parallel to the local magnetic field. The problem then consists in solving Eq. (21) under the conditions of a bouncing test electron. This task resembles the solution under non-magnetised conditions which had been given in our previous paper (Treumann & Baumjohann, 2014). In the known form it cannot be applied here but has to be substantially modified in order to become adapted to the conditions of electron trapping in mirror modes.

## 3.1 Conditions for an attractive potential

In the light of the previous discussion we rewrite Eq. (21) in the magnetic field as

$$\begin{aligned}
\Phi(\boldsymbol{x},t) &= -\frac{e\lambda_e^2}{2(2\pi)^2\epsilon_0} \int \frac{\omega_{\boldsymbol{k}}\,d\omega\,d\boldsymbol{k}\,e^{i\boldsymbol{k}\cdot(\boldsymbol{x}-\boldsymbol{v}t)}}{(1+k_\perp^2\lambda_e^2+k_\|^2\lambda_e^2)} \times \\
&\quad \times \left[\frac{\delta(\omega-k_\|v_\|)}{(\omega-\omega_{\boldsymbol{k}})} - \frac{\delta(\omega-k_\|v_\|)}{(\omega+\omega_{\boldsymbol{k}})}\right]
\end{aligned} \qquad (24)$$

Here we left the Debye-potential term out as it is of no interest, and resolved the denominator. We also refer to the parallel particle velocity $v_\| = v\cos\theta$ which in our case of mirror trapped test particles is along the magnetic field. It selects the parallel wavenumber of the wave in the Dirac $\delta$-function to replace the frequency $\omega$. In the same spirit the argument of the exponential becomes $i\boldsymbol{k}\cdot(\boldsymbol{x}-\boldsymbol{v}t) = ik_\perp\rho\sin\phi+ik_\|(s-tv\cos\theta)$ with $\rho$ the independent perpendicular spatial coordinate. It is assumed that the magnitude $v$ of the velocity remains constant in this kind of interaction, which holds for the adiabatic motions along the magnetic field where no further external force acts on the electron except for the stationary restoring magnetic force. (Note also that the wave frequency $\omega_{\boldsymbol{k}}$ depends on $k_\|, k_\perp$ but not anymore on angle $\phi$ because it has been determined independently from kinetic wave theory not using the test particle picture.)

These assumptions reduce the integral to integrations over the perpendicular wavenumber $k_\perp, \phi$, and frequency $\omega$. Moreover, since the problem has become cylindrically symmetric with respect to $\boldsymbol{B}$, integration over $\phi$ can easily be performed by using the representation of the exponential as a series of Bessel functions (Gradshteyn & Ryzhik, 1965) which reduces to the zero-order Bessel function $J_0(k_\perp\rho)$. The formal result before final integration is

$$\begin{aligned}
\Phi(s,\rho,t) &= -\frac{e\lambda_e^2}{2(2\pi)^2\epsilon_0} \int \frac{\omega_{\boldsymbol{k}}\,d\omega\,k_\perp dk_\perp\,dk_\|}{(1+k_\perp^2\lambda_e^2+k_\|^2\lambda_e^2)} \times \\
&\quad \times \left[\frac{\delta(\omega-k_\|v_\|)}{\omega-\omega_k} - \frac{\delta(\omega-k_\|v_\|)}{\omega+\omega_k}\right] \times \\
&\quad \times J_0(k_\perp\rho)\,e^{ik_\|(s-v_\|t)}
\end{aligned} \qquad (25)$$

where one understands $v_\| = v\cos\theta(s)$. We note again that this form is still valid also for electromagnetic waves if only the frequency is understood as the solution of the electromagnetic dispersion relation. In view of later application to the mirror mode we now restrict to purely electrostatic waves, in our case ion sound which, in contrast to other electrostatic waves like Bernstein modes, has the right property of propagating along the magnetic field with parallel electric field. The ion sound wave frequency is

$$\omega_k^2 \approx \frac{\Lambda_0(\eta_i)c_s^2 k_\|^2}{\Lambda_0(\eta_e)+k^2\lambda_e^2} \approx \frac{\Lambda_0(\eta_i)k_\|^2 c_s^2}{1+k^2\lambda_e^2} \qquad (26)$$

with the right-hand side holding since the electron term in the denominator is $\Lambda_0(\eta_e) \approx 1$. In the low frequency approximation applicable here, the frequency is proportional to the parallel wave number. In the following we simplify this dispersion relation setting $\Lambda_0(\eta_i) \approx 1$, which is its maximum value, and in the resonant denominators neglecting the inverse dependence of $\omega_k$ on $k\lambda_e$, only keeping it in the nominator of the integral. Then one may perform the integration with respect to $k_\perp$ which gives, with $\xi = \lambda_e k_\perp$, $\rho' = \rho/\lambda_e$, $\zeta = k_\|\lambda_e$,

$$I(\rho',\zeta) \equiv \int_0^\infty \frac{\xi\,d\xi\,J_0(\xi\rho')}{\left(1+\zeta^2+\xi^2\right)^{3/2}} = \frac{\exp\left(-\rho'\sqrt{1+\zeta^2}\right)}{\sqrt{1+\zeta^2}} \qquad (27)$$

In order to perform the integral, its singular properties have to be elucidated. The dominant contribution will come from the resonant denominators in the bracketed terms. Any possible resonances in the Coulomb factor do not play any role here. The Dirac $\delta$-functions prescribe replacing the frequency everywhere with $k_\|v_\|$. It is, however, convenient to delay this action until integrating out the singularities in the complex $\omega$ plane. To see their effect, one temporarily replaces $k_\|$ in the argument of the exponential with $\omega$ as the $\delta$-function prescribes as an inverse action. Then we have for $ik_\|(s-v_\|t) = i\omega(s/v_\|-t)$. Since the waves are damped, the imaginary part of the frequency is required to be negative. This forces demanding $s/v_\| - t < 0$, consequently taking the $\omega$-integration over the lower complex $\omega$-half plane, which in surrounding the poles in the positive sense adds a factor $2\pi i$ to the integral and includes the sum of residua $\omega = \pm\omega_k$ in

the integral in this order. The result is

$$\Phi(s,\rho,t) = -\frac{ie}{4\pi\epsilon_0\lambda_e}\frac{c_s}{v_\parallel}\int \frac{\zeta\, d\zeta\, e^{-\rho'\sqrt{1+\zeta^2}+i\zeta(s-v_\parallel t)/\lambda_e}}{\sqrt{1+\zeta^2}}$$

$$\times \left[\delta\left(\zeta-\frac{\omega_k\lambda_e}{v_\parallel}\right)-\delta\left(\zeta+\frac{\omega_k\lambda_e}{v_\parallel}\right)\right] \quad (28)$$

Performing the substitution prescribed by the delta functions in the exponential only yields the sum of two exponentials which turns into a sine function. One then obtains for the potential of the particle in the presence of ion sound waves

$$\Phi(s,\rho,t) = \frac{e}{2\pi\epsilon_0\lambda_e}\frac{c_s}{v_\parallel}\int_0^1 \frac{\zeta\, d\zeta\, e^{-\rho'\sqrt{1+\zeta^2}}}{\sqrt{1+\zeta^2}}$$

$$\times \sin\left[\zeta(s-v_\parallel t)/\lambda_e\right] \quad (29)$$

What remains is the $\zeta$ integration with $\zeta=k_\parallel\lambda_e<1$ limited. To simplify, we can either neglect $\zeta$ or replace it by unity in the arguments of the roots. To be conservative and decide for the weakest case, we chose the latter, what yields the integral

$$\Phi(s,\rho,t) = -\frac{e}{4\sqrt{2}\pi\epsilon_0\lambda_e}\frac{c_s}{v_\parallel}e^{-\sqrt{2}\rho'}\times$$

$$\times \int_0^1 d\zeta^2\sin\left[\zeta|s-v_\parallel t|/\lambda_e\right] \quad (30)$$

The argument of the sine function is negative. So we have taken its sign out and use its absolute value. Integration gives

$$\Phi(s,\rho,t) = -\frac{e}{2\sqrt{2}\pi\epsilon_0}\frac{c_s}{v_\parallel}\frac{e^{-\sqrt{2}\rho/\lambda_e}}{|\sigma|^2}\times$$

$$\times\left\{\sin|\sigma|-|\sigma|\cos|\sigma|\right\} \quad (31)$$

where

$$\sigma = (v_\parallel t-s)\lambda_e^{-1}>0$$

The condition for an attractive potential follows immediately as

$$\tan|\sigma|>|\sigma| \quad \text{or} \quad 0<\sigma<\frac{\pi}{2} \mod (2\pi) \quad (32)$$

Depending on the parallel velocity $v_\parallel>0$ there is an entire range of distances $s<v_\parallel t<\pi\lambda_e/2$ in which the conditions for an attractive potential are satisfied. We may note that for negative velocities $v_\parallel<0$ there is no range where the potential can become attractive as the braced expression is always positive. It is the scalar product $\boldsymbol{k}\cdot\boldsymbol{v}$ between the wave number of the ion-sound and the test particle velocity which selects those speeds which are parallel to the sound velocity, not anti-parallel. One should keep in mind that this attraction has nothing in common with wave trapping, however! It is the over-screening effect of the particle, which is moving on the background of the wave noise and experiences the modified dielectric properties of the plasma.

It should also be noted that in this condition the time explicitly appears because the test electron is seen from the stationary observers frame in which the electron moves. Instead, $\sigma$ is measured in the moving electron frame. This distinction is important to make as it will be picked up again below.

The restriction on the velocity is obtained from that $\omega^2<\omega_k^2$ when referring to the replacement $\omega=k_\parallel v_\parallel=k_\parallel v\cos\theta$ prescribed by the $\delta$-functions. Rescaling $\omega_k\sim c_s k_\parallel$, it follows that the parallel particle speed is limited as

$$|v_\parallel|\lesssim c_s \quad \text{or} \quad |\cos\theta|\lesssim\frac{c_s}{v} \quad (33)$$

This is in fact a condition on the angle $\theta$. For small speeds $v<c_s$ the condition is trivial. The largest effect is caused when the particle speed is parallel, below and close to the phase velocity $c_s$ of the ion-sound wave. For large velocities $v>c_s$ the angle between the phase speed and velocity must be close to $\pi/2$, in agreement with the above requirement on the potential becoming attractive.

This is an important point in application to a plasma. In thermal plasmas we have generally $c_s\approx\sqrt{m_e/m_i}\,v_e$ which is far below the thermal speed. Hence there are only few electrons in the distribution sufficiently far below thermal speed which would satisfy the resonance condition $v<c_s$. Higher speed electrons can be in resonance and thus contribute to attraction only at strongly oblique wave and electron speeds. Consequently under normal conditions in a plasma the generation of attractive potentials becomes obsolete, a point which had been missed in previous work (Neufeld & Ritchie, 1955; Nambu & Akama, 1985). In the particular case of mirror modes it becomes the crucial ingredient, as will be demonstrated below.

### 3.2 Correlation length

In all cases the attraction exceeds the repulsion *outside* the Debye sphere of the electron in its wake and, therefore and most important, can be felt by other electrons. From here it is clear that two electrons must move at distance somewhat larger than $\lambda_e$ and at nearly same speed in the same direction in order to be held together by their attractions and form a pair. This is the important point when applying our model to the mirror mode below.

Having obtained the conditions under that the wake potential behind the moving test electron becomes attractive, we would like to know the distance over that the negative potential extends. This distance is measured in the instantaneous frame of the electron and is, hence, given by the above absolute normalised value of $|\sigma|<\pi/2$ which repeats itself periodically. It is, however, clear that it is only the zeroth period which counts as the effect of the dielectric polarisation on the electron diminishes with increasing distance $s'=\sigma\lambda_e$. In absolute numbers this distance becomes

$$\lambda_{corr}=|s-v_\parallel t|<\frac{\pi}{2}\lambda_e\approx 1.57\lambda_e \quad (34)$$

which can be understood as an electron "correlation length" between neighboured electrons. Any electrons within such a distance will behave about coherently, an important conclusion which, however, has to be extended below to many electrons.

This correlation length is to be compared with the particle spacing in the plasma. Plasmas are defined for particle densities $N\lambda_e^3 \gg 1$, which implies that the distance between the particles is $\ll \lambda_e$. Consequently the extension of the attractive potential in the electron wake is much larger than the spatial distance between two electrons. It thus affects many electrons, an effect which cannot be neglected when speaking about attraction.

As for an example, in the magnetosheath which is the preferred domain where the mirror mode is permanently excited, the average density is, say, $N \approx 3 \times 10^7 \text{ m}^{-3}$ at temperature $T_e \approx 50 - 100$ eV. For the Debye length we have $\lambda_e \approx 10$ m, while the inter-particle distance is a mere of $\approx 0.005$ m. Roughly $\approx 10^4$ electrons should experience the presence of the attraction behind the test particle, which thus becomes a many-electron effect. Because pair formation depends on the quite severe condition on the particle velocity, not all those electrons of course will form pairs though however, in reality, the attractive potential involves a substantial fraction of electrons which necessarily will cause modifications of the plasma conditions. Normally such modifications will only cause minor effects in the wave spectrum and will be negligible. Below we show that in the evolution of the mirror mode they become important.

### 3.3 Ensemble averaged potential

If we understand the plasma as a compound of a large number of electrons, we can ask for the ensemble averaged potential $\langle \Phi \rangle$ of the single electron averaging over the particle energy distribution. In an isotropic plasma this is the Boltzmann distribution. Writing for the parallel velocity $v_\parallel = v\cos\theta$ the average potential becomes

$$
\begin{aligned}
\langle \Phi \rangle \;=\; & \frac{e}{\epsilon_0}\frac{Cc_s}{\sqrt{2}\lambda_e}e^{-\sqrt{2}\rho/\lambda_e}\int_0^\infty vdv\,e^{-v^2/v_e^2} \times \\
& \times \int_{s/tv}^{(s+\pi\lambda_e/2)/tv} \frac{d\cos\theta}{\sigma^2\cos\theta}\big[\sin\sigma - \sigma\cos\sigma\big]
\end{aligned}
\tag{35}
$$

which immediately tells that the mean potential taken over the full Boltzmann distribution in repulsive. This is clear, however, because it accounts for all electrons in the Debye sphere. To calculate the cos-integral we expand the trigonometric functions to obtain

$$
\frac{\pi}{6}\int \sigma d\cos\theta \approx \frac{\pi}{6\lambda_e}\left[\frac{\pi^2}{12}vt - s\log\left(1 + \frac{\pi}{2}\frac{vt}{s}\right)\right]
\tag{36}
$$

We now exclude the Debye sphere by restricting the integration with respect to $v$ over a shell between the thermal and trapped speeds. This gives

$$
\begin{aligned}
\int_{T_e}^{\mathcal{E}_{trap}} d\mathcal{E}\,e^{-\mathcal{E}/T_e}&\left[\frac{\pi^2 t}{12}\sqrt{\frac{2\mathcal{E}}{m_e}} - s\log\left(1 + \frac{\pi t}{2s}\sqrt{\frac{2\mathcal{E}}{m_e}}\right)\right] \approx \\
& -\left(1 - \frac{\pi}{6}\right)\frac{\pi t}{2}\sqrt{\frac{2T_e^3}{m_e}}\int_1^y x^{\frac{1}{2}}e^{-x}dx
\end{aligned}
\tag{37}
$$

For a mean attractive potential the last integral should be positive. Doing it yields (Gradshteyn & Ryzhik, 1965)

$$
\int_1^y x^{\frac{1}{2}}e^{-x}dx = \frac{2}{3}\left(y^2 e^{-y} - e^{-1}\right) \approx -y^3 + 2y^2 - 1
\tag{38}
$$

which is positive only if $y = 1 + \Delta$ and $\Delta = (\mathcal{E}_{trap} - T_e)/T_e < 1$ in which case there is a narrow energy range (or energy "gap") for trapped electrons where the mean potential $\langle \Phi \rangle < 0$ becomes attractive for the electrons when averaging over their energy distribution and warranting that they behave coherently. The latter we will show can under certain condition be the case.

### 4 Two-electron potential

We saw that, under a certain condition, an electron moving in the plasma in resonant interaction with an ion-sound background may give rise to an attractive potential in its wake where another electron can be captured and thus be forced to accompany the first electron. First of all, in plasma all electrons are in permanent motion. Hence, if an electron satisfies the resonance condition with an ion sound fluctuation, it acts attracting on another one moving nearly at same speed. We have seen that this attractive potential in the presence of a large number of thermally distributed electrons becomes depleted. This holds when just one electron contributes to the potential. We now extend this to the combined effect of two electrons in the interaction, in which case we can immediately use the above solution when, however, accounting for the slightly different velocities $v_{\parallel 1}, v_{\parallel 2}$ and initial locations $s_1, s_2$ of the electrons along the magnetic field. In view of the later application to mirror modes, we again consider only motion along the magnetic field not yet specifying to the peculiarities introduced by bouncing in the mirror field. Then the two-electron potential becomes

$$
\begin{aligned}
\Phi(s,\rho,t) = \; & - \sum_j \frac{e}{2\sqrt{2}\pi\epsilon_0}\frac{c_s}{v_{\parallel j}}\frac{e^{-\sqrt{2}\rho/\lambda_e}}{|\sigma_j|^2} \times \\
& \times \left\{\sin|\sigma_j| - |\sigma_j|\cos|\sigma_j|\right\}
\end{aligned}
\tag{39}
$$

with $j = 1, 2$ counting the electrons. Here

$$
\sigma_j = (v_{\parallel j}t - s_j)\lambda_e^{-1} > 0
$$

As before, the requirement $\sigma_j > 0$ results from the condition that the waves in resonance with the electrons must be damped. In order to obtain the combined effect of the two electrons, we transform to their centre-of-mass frame

$$2Z = s_1 + s_2, \qquad 2z = s_1 - s_2$$
$$2U = v_{\|1} + v_{\|2}, \qquad 2u = v_{\|1} - v_{\|2} \tag{40}$$

From the previous we saw that the large correlation length implies that many electrons are affected. Any attractive potential couples two ore more particles together. The most probable state to be formed is the two-particle (singlet) state. These will be distributed over the plasma, resembling the Cooper states in solid state superconductivity while not being a quantum effect here. Rather it is the polarisation effect moving particles produce in the high temperature collisionless plasma which causes singlet states of pairs.

In order to be realistic, we now derive the condition for singlet states to evolve. To simplify the algebra, let us define

$$\Sigma \ :=: \ \tfrac{1}{2}\big(\sigma_1 + \sigma_2\big) \quad \equiv \ \big(Ut - Z\big)\lambda_e^{-1} \ > 0$$
$$\sigma' \ :=: \ \tfrac{1}{2}\big(\sigma_1 - \sigma_2\big)\lambda_e \ \equiv \ \big(ut - z\big)\lambda_e^{-1} \tag{41}$$

The restriction on $\Sigma > 0$ maps the $\omega$-resonance onto the new variables. At the contrary, $\sigma'$ can be positive or negative. With these expressions and after some rather tedious though simple calculations, Eq. (39) can be brought into the form

$$\Phi(s, \rho, t) \approx \ - \ \frac{\sqrt{2}e}{\pi \epsilon_0} \frac{c_s t}{(\lambda_e \Sigma + Z)} \frac{e^{-\sqrt{2}\rho/\lambda_e}}{|\Sigma|^2} \times$$
$$\times \ \Big\{ \sin|\Sigma| - |\Sigma|\cos|\Sigma| \Big\} \cos\sigma' \tag{42}$$

where we made use of the above representations and replaced

$$v_{\|1,2}t = \tfrac{1}{2}\lambda_e\big(\Sigma \pm \sigma'\big) + Z \pm z \tag{43}$$

This expression for the potential holds under the reasonable assumptions $\sigma' \ll \Sigma$ and $z \ll Z$ that the difference between the two electrons in location is small enough to be found within the correlation length.

Only under this condition one expects that the electrons will be correlated. Interestingly, the form of the potential remains the same as that for the one-particle case with the only difference that the potential is multiplied by $\cos\sigma'$. Hence the condition for attraction depends on the value of $\sigma'$.

Closely spaced electrons of similar and, as required, resonant speeds not differing too much from the phase speed of the ion-sound give indeed rise to attraction between the two electrons if the following conditions are satisfied:

$$\tan\Sigma \ > \ \Sigma \qquad \text{if} \qquad \cos\sigma' > 0$$
$$\tan\Sigma \ < \ \Sigma \qquad \text{if} \qquad \cos\sigma' < 0 \tag{44}$$

which yields

$$0 \ < \ \Sigma \ < \ \frac{\pi}{2} \qquad \text{if} \qquad \cos\sigma' > 0$$
$$-\frac{\pi}{2} \ < \ \Sigma \ < \ 0 \qquad \text{if} \qquad \cos\sigma' < 0 \tag{45}$$

These conditions are essentially the same as in the one-electron case. There modification is due to $\cos\sigma'$ being positive or negative and that they apply to the centre of mass coordinate system $Z$ and mean particle speed $U$ which both are contained in the variable $\Sigma$.

We remark that these conditions are very general. They substantially generalise the conditions found earlier by Nambu & Akama (1985) to the much more important interaction between two electrons, the lowest order singlet state and thus most realised state in a plasma. Actually, the attractive potential of one single electron makes little sense as it has an effect only if it affects another electron. This is exactly what happens in the case of Cooper pairs where the attraction becomes important only in an assembly of many electrons, as was realised in BCS theory (Bardeen et al., 1957). In the same vain we are proceeding here.

Higher order states like interaction of three electrons leading to triplets and so on are in principle also possible but will not play any important role, because the interaction decays with distance, even though they may be located within the correlation length and form "quasi-particles". In the singlet state, the paired electrons behave like one particle of double charge and double mass for the time of their interaction, the time they remain inside one correlation length. This length for the singlet is the same as was given above, produced by one electron, with the only difference that it now applies to the centre of mass of the two electrons. Measured from the centre of mass it extends to its both sides over a length of roughly $\lambda_{corr} \approx 1.5\lambda_e$.

In physical units the first singlet state, for instance, is realised for

$$0 < Ut - Z < \frac{\pi}{2}\lambda_e, \qquad |ut - z| \ll \frac{\pi}{2}\lambda_e \tag{46}$$

In these cases resonant electrons, in the presence of an ion-sound wave background, will arrange into loosely bound electron pairs. In high temperature plasmas, a substantial number of such pairs will exist. However, they will mostly not play any role in the dynamics. In order to do so, the plasma must offer additional ways for the bound singlet pair states to cause any susceptible effect in the plasma. Such conditions are provided by the quasilinearly stable mirror mode and will be exploited below.

## 5 Mirror bottle and pairs

Being in the possession of the conditions under which electron pairs can form in a high temperature plasma in interaction with a thermal background of ion sound waves, we now

intend to apply them to the case of mirror modes. We saw that the correlation length between electrons provided by one single test electrons is of the order of $\lambda_{corr} \sim 1.5\lambda_e$. This value is only slightly increased by the interaction of two electrons, such that we can roughly take $\lambda_{corr} \sim 2\lambda_e$ for the singlet.

A mirror bottle is a preferred place for pair formation. This is in contrast to any spatially extended plasma. Firstly, the bottle confines trapped electrons which cannot easily escape. Secondly, the parallel velocity of a bouncing electrons varies along the mirror magnetic field and at some place may get in resonance with the thermal ion-sound spectrum present in the entire plasma volume. If this happens at some location along the mirror magnetic field, electrons might form pairs and remain correlated for some time, bunch, perform like bound orbits and thus represent resonant correlated states which due to the correlation become coherent states.

### 5.1 Centre of mass pair bounce motion

The application of these finding to mirror modes is not an easy task. The electrons perform a complicated bounce motion along the inhomogeneous magnetic field with periodically changing bounce velocity and bounce frequency depending on the value of their constant magnetic moment. Under these conditions we need to know the variation of their bounce velocity as function of the location along the magnetic field between the mirror points.

We moreover need to satisfy the common resonance condition of the pairs with respect to the phase velocity of the ion sound. Since the electron velocity is generally much larger than the latter this immediately suggests that the best conditions for attraction will be found near the mirror points $s_m$. There the parallel velocity of the electrons drops to zero, and there will be a certain range $\Delta s$ at distance $s \lesssim s_m$ where the resonance condition is satisfied most easily. Near $s_m$ one expects that attraction will become important.

In order to understand this process we thus need to transform to the moving frame of the pairing electrons. For this purpose we use the electron bounce motion to define the new pair-electron quantities

$$\mathcal{M} =: \tfrac{1}{2}\big(\mu_1 + \mu_2\big), \qquad \mu =: \tfrac{1}{2}\big(\mu_1 - \mu_2\big) \tag{47}$$

$$\Omega^2 =: \mathcal{M}B_0''/m_e, \qquad \varpi^2 =: \mu B_0''/m_e \tag{48}$$

$$U^2 =: \frac{2}{m}\mathcal{E} - \Omega^2 Z^2, \qquad \mathcal{E} =: \tfrac{1}{2}\big(\mathcal{E}_1 + \mathcal{E}_2 - 2\mathcal{M}B_0\big) \tag{49}$$

$$u^2 =: \frac{2}{m}\tilde{\epsilon} - \varpi^2 z^2, \qquad \tilde{\epsilon} =: \tfrac{1}{2}\big(\mathcal{E}_1 - \mathcal{E}_2 - 2\mu B_0\big) \tag{50}$$

The mean bounce velocity $U$ of the pair becomes a function of the location $Z$ of the centre of mass along the magnetic field. This requires knowledge of its displacement as a function of the bounce phase which again requires solution of the two dynamics of the two electrons. Note the adiabatic constants $\mathcal{E}, \mathcal{M}, \mu, \Omega, \varpi$. The only variables are the mean and difference velocities $U(Z), u(z)$.

In the magnetic mirror symmetry, $U(t)$ is the bounce velocity of the trapped electron pair, and $Z(t)$ is its location along the magnetic field at time $t$. The difference speed $u(z)$ is measured in the centre of mass frame relative to $Z$ and $U$.

The mean speed $U$ along the magnetic field must be expressed either as function of time $t$ or distance $Z$. For this to accomplish one needs to solve the parallel equation of motion:

$$\frac{dU}{dt} = -\Omega^2 Z - \varpi^2 z \approx -\Omega^2 Z, \qquad U = \frac{dZ}{dt} \tag{51}$$

which is given in the reasonable approximation of small $\varpi^2 z$. Obviously the mean speed along the field obeys the mean bounce equation, an oscillation at frequency $\Omega$. Integrating the bounce equation of motion with $U(Z) = \sqrt{2\mathcal{E}/m - \Omega^2 Z^2}$ yields

$$Z(t) = Z_m \sin\left(\frac{\pi t}{2t_m}\right), \qquad Z_m = \Omega\sqrt{\frac{2\mathcal{E}}{m_e}} \tag{52}$$

$Z_m$ is the distance of the centre of mass mirror point reached by the pair at mirror time $\Omega t_m = \tfrac{1}{2}\pi$ along the magnetic field. Symmetric mirror bottles have been assumed, implying time symmetry $\pm t_m$.

For the lag in distance $z$, as measured relative to the centre of mass $Z$, we obtain similarly

$$z(t) = z_\epsilon \sin\left(\frac{\pi t}{2t_\epsilon}\right), \qquad z_\epsilon = \varpi\sqrt{\frac{2\tilde{\epsilon}}{m_e}} \tag{53}$$

with $\varpi t_\epsilon = \tfrac{1}{2}\pi$ the lag in time in the electron pair to reach the mirror point at relative location $z_m$ to the mirror location of the centre of mass $Z_m$.

These expressions give the centre of mass and jitter velocities as functions of time

$$U(t) = \frac{\pi}{2}\frac{Z_m}{t_m}\cos\left(\frac{\pi t}{2t_m}\right) \tag{54}$$

$$u(t) = \frac{\pi}{2}\frac{z_\epsilon}{t_\epsilon}\cos\left(\frac{\pi t}{2t_\epsilon}\right) \tag{55}$$

It is important to remark that at this point we have only transformed to the centre of mass motion. We have not yet determined the paring conditions. The first of the former expressions thus gives the two-electron bounce velocity in the centre of mass frame, the second the jitter velocity around this centre.

In order to apply to the quasilinear stable mirror mode, one must transform to the bounce motion. This can easily be done inside the mirror bottle by using the parallel equation of motion and the magnetic trapping conditions. It implies substituting the parallel bounce solution for $t(s)$. We shall do it later below. Instead, here, it is first more important to obtain the condition of pair formation in the two-electron potential.

## 5.2 Condition for pair formation

Electron pair formation proceeds if, in addition to the conditions for attraction which have been given above, the pair electron are in resonance with the ion sound. This condition is non-trivial. We mentioned already that electrons participating in attraction move at speed comparable to the thermal speed $v_e$ which exceeds $c_s$ substantially. Under non-mirror conditions pair formation will thus barely take place. However, magnetic mirrors as provided by the mirror instability are a rare exception.

The resonance condition is in fact not a condition on $U(Z)$ but on the angle between the pair velocity and the direction of the magnetic field, as the latter is the direction of the propagation of the ion sound. During the bounce motion the particle velocities are adiabatically conserved. It is only the angle $\theta(s)$ that changes along the magnetic field. Thus writing $U(s) = \frac{1}{2}v\big(\cos\theta_1(s) + \cos\theta_2(s)\big)$, assuming that $v_1 \approx v_2, \mu_1 \approx \mu_2$, we have

$$\cos\theta_1 = \frac{U+u}{v}, \qquad \cos\theta_2 = \frac{U-u}{v} \qquad (56)$$

Introducing the mean angles $\Theta = \frac{1}{2}(\theta_1 + \theta_2), \vartheta = \frac{1}{2}(\theta_1 - \theta_2) \ll \Theta$ we obtain

$$\begin{aligned}
\frac{\langle U \rangle}{v} &= \cos\frac{\Theta}{2}\cos\frac{\vartheta}{2} \approx \cos\frac{\Theta}{2} \\
\frac{\langle u \rangle}{v} &= -\sin\frac{\Theta}{2}\sin\frac{\vartheta}{2} \approx -\frac{\vartheta}{2}\sin\frac{\Theta}{2}
\end{aligned} \qquad (57)$$

Note that $u$ can be negative as it is measured in the centre of mass frame. The condition of resonance $U(Z) \lesssim c_s$ then reduces to

$$\begin{aligned}
\langle U \rangle &\lesssim c_s &\longrightarrow&& \cos\frac{\Theta}{2} &\lesssim \frac{c_s}{v} \ll 1 \\
\left|\frac{\vartheta}{2}\right| &\sim &\frac{\langle u \rangle}{v}& \ll 1
\end{aligned} \qquad (58)$$

This condition shows that our assumption of about equal magnitudes $v_1 \approx v_2$ is not crucial because of the smallness of this ratio. It shows moreover that the resonance condition is nicely satisfied near the mirror points $s_m$, where the average angle $\Theta/2 \approx \frac{\pi}{2}$.

As for an example, $c_s/v_e \approx \sqrt{m_e/m_i} \approx 0.023$ which shows that the average cosine is very small, and the effective angle $\Theta/2 \approx 88.7°$ is close to $90°$. Allowing for a deviation in the ratio of $\sim 0.002$ the angular variation would amount to $\vartheta \approx 0.2°$ as obtained from the average jitter velocity $\langle u \rangle$, as is suggested by the second above condition with $\sin\Theta/2 \approx 1$, which gives the angular spread in case of attraction.

Once a mirror bottle evolves, there is a narrow spatial range near the mirror points $\pm s_m$ along the magnetic field for the trapped electrons to generate attractive potentials in their wake during their bounce motion inside the magnetic mirror trap. This is the range of resonance where the parallel speed matches the slow speed of ion sound. This attractive potential extends over approximately one to two Debye lengths along the magnetic field outside the Debye sphere of the acting electrons (roughly some ten meters in the magnetosheath!) whose charge fields are compensated by the bulk of the surrounding electrons populating the Debye sphere.

This length is however much larger than the mutual particle distance. It thus affects a substantial number of electrons which, in case their velocities do not differ much, form pairs within a correlation length which the attractive potential attributes to them.

As a consequence, there is a substantial number of paired electrons inside the mirror bottle along the magnetic field around all the many mirror points of trapped electrons of different initial angle and velocity. The distribution of those mirror points depends on the (equatorial) pitch angle distribution of the electrons trapped in the field minimum $B = B_0$ at the centre of the mirror bottle. One thus expects that over a certain length along the mirror magnetic field an almost homogeneous distribution of electron pairs will evolve.

The attractive potential acts to combine two electrons into a pair at location close to $Z = s_m$, where the centre of mass velocity of the electrons goes into resonance with $U \approx c_s \ll v_e$. This effect attaches the pair to the particular group of ion sound waves in resonance with the pair and thus locks the pair to them.

In addition there is a small jitter velocity $u$ around the resonance left over which is required to be small in order not to destroy the resonance. Investigation of the stability of pairs in this case is a separate problem which we do not attack in this communication, even though it is important for the further considerations.

However, we may note that even if pairs will not be stable for long time, not becoming locked completely to the group of ion sound waves with resonant wave numbers, they will stay for some finite time in resonance and will always be replaced by other, newly formed, pairs when dispersing. Since the electrons and pairs are all completely indistinguishable, a fluctuating population of pairs will thus always be present, if only the conditions for pairing exist. In this sense, a mirror mode is the ideal and possibly the only place in high temperature plasma physics, where pairing can and probably will occur. We may also point on a similar mechanism for ions. Our basic expression for the potential for ions would be similar to that for the electrons. It moreover holds for any kind of waves while pairing in addition requires that the particle candidates for pairing must be capable of resonating with the waves. This latter condition is rather severe and therefore the necessary condition to have pairing.

## 5.3 Dynamics of pair population

So far we just derived the conditions for pairing in the centre of mass system of the bouncing electrons. It is most interesting what happens after pairing. Naively thinking nothing. Pairs form and may dissociate. Those formed and do not disintegrate, may perform a combined bounce motion as pre-

scribed by the centre of mass bounce equation given above. In this case, they remain pairs but bounce between their centre of mass mirror points. Thus, for some pairs, both disintegration and common bounce may be the case.

In addition there will be a number of pairs, who are stable, at least for some longer time. They neither disintegrate nor bounce, because they remain stable at resonance with the group of phonons of resonant wave numbers $k$ near $Z \approx s_m$ and centre of mass speed $U \approx c_s$. As long as this interaction holds they will not be able to return into bouncing simply, because the phonons, which are independent of the particle motion, not being subject to bounce or any mirror force, do not allow the paired electrons to return into bouncing. These phonons continue their slow motion along the magnetic field and, in this way, lock the pair.

Vice versa, by locking the pair, they become themselves locked near the pair's mirror point. Thus such pairs drop out of bouncing and, for their life time, become locked near $s_m$. The length of this time is a question of their stability in which not only the two electrons, but also the ion sound phonons are involved, while these are independent of the bounce. Stable pairs still posses a small jitter speed $u$ around $s_m$, which is insufficient for re-injecting them into the bounce. They move together with the phonons further up the field at reduced parallel speed $\sim c_s$.

The condition on the jitter energy $m|u|^2$ for stability is that it should not exceed the trapping potential $\frac{1}{2}m^*u^2 \lesssim e|\Phi|$. Pairs of smaller jitter energy should thus be stable. A more precise selection rule requires the solution of the stability problem.

We consider a mirror bottle. The pitch angle distribution of trapped electrons in a mirror bottle is not known a priori. The equatorial pitch angle $\theta_0$ is given as

$$\sin^2 \theta_0 = B_0/B(s_m) \qquad (59)$$

the ratio of minimum magnetic field to the mirror field of trapped electrons. Electrons with large equatorial pitch angle mirror very close to the minimum magnetic field. Electrons with small equatorial pitch angle mirror near the end of the bottle. It is thus clear that there is practically a continuous distribution of mirror points along the mirror magnetic field in the bottle depending on the given initial distribution of equatorial pitch angles. Moreover this applies to all magnetic field lines inside the bottle, not only the central one.

The dependence of the mirror points $s_m$ on electron velocity $v < \sqrt{\mathcal{E}_{tr}/m_e}$, location $s$, and pitch angle $\theta_0$ is obtained from the bounce frequency $\omega_b$, the location along the field

$$s(t) = s_m \sin \omega_b t \qquad (60)$$

and the time to reach from $s = 0$ to $s$ within the bounce motion

$$t(s) = \int_0^s \frac{ds/\cos\theta_0}{\sqrt{1 - s^2(B_0''/2B_0)\tan^2\theta_0}} \qquad (61)$$

Solving the latter integral and resolving for the mirror point yields the wanted expression

$$\begin{aligned} s_m(v,s,\theta_0) &= \frac{s}{\sin\left[\eta(s,v,\theta_0)\right]} \\ \eta(s,v,\theta_0) &\equiv v\sin^{-1}\left(s\sqrt{\tfrac{1}{2}B_0''/B_0}\,\tan\theta_0\right) \end{aligned} \qquad (62)$$

This is a complicated though continuous dependence on $s$ and $\theta_0$. The distribution of mirror points on the mirroring electron velocity respectively its energy can, in principle, be integrated over the range of velocities $0 < v - v_e < v_{trap} - v_e$ contributing to pairs. Assuming an equatorial isotropic gaussian-velocity distribution $\sim f(\theta_0)\exp(-\epsilon/T_e)$ yields

$$\langle s_m(s)\rangle \propto \left(\frac{\Delta\mathcal{E}_{pair}}{T_e}\right) \int_0^{\pi/2} \frac{s\,d\theta_0\,\sin\theta_0\,f(\theta_0)}{\sin^{-1}\left(s\sqrt{B_0''/B_0}\,\tan\theta_0\right)} \qquad (63)$$

with $\Delta\mathcal{E}_{pair} = \mathcal{E}_{trap} - T_e > 0$. For any given equatorial pitch angle distribution $f(\theta_0)$ there is a corresponding distribution of mirror points inside the mirror bottle along $s$, i.e. along the magnetic field $\boldsymbol{B}(s)$.

As a consequence, the entire narrow volume of the quasi-linearly stable mirror bottle will be subject to the presence of pairs, each of which is located at and along the magnetic field line centred around its common pair-mirror point. (One may note that the point $s = 0$ does not contribute to the integral, neither mathematically nor physically because it is not a real mirror point. Also $\theta_0 = 0$ does not contribute because those particles do not participate in mirroring.)

Under these conditions the pairs become an important part of the population of a mirror bottle and might contribute to its dynamics. The locked pairs have twice the electron mass $m^* = 2m_e$ and twice the electron charge $q^* = -2e$. Energetically, their parallel motion condensates in the lowest energy bounce level $\tilde\epsilon \sim (m^*/2)\langle u^2\rangle \ll \mathcal{E}_{trap}$, the energy in their average jitter motion around the mirror point, negligibly small with respect to their gyration energy which, near the mirror point, has absorbed almost all kinetic energy into the gyration.

At same magnetic field strength $B(s_m)$ they thus have equal gyroradii, concentrating in a current shell which represents a surface current $J_{\perp pair} = -e^* N_{pair} v_e \Delta r$ on the timescale of the bounce, which averages over all particle phases. This shell has width $\Delta r \sim (v_e/\omega_{ce})\Delta B/B$. The surface current it carries might give rise to an integrated localised orbital diamagnetism. In the last subsection below we explore its effect.

## 5.4 Pair induced mirror growth

The promising macroscopic effect is the direct contribution of pairs to both, the evolution of an electron anisotropy and its effect on the ion mirror mode, which is the content of the following subsections.

A gyrating locked pair population has large perpendicular energy $\sim 2T_e$ and very small parallel energy $\sim m_e|u|^2$. It hence possesses a large anisotropy

$$A_{pair} = \frac{2T_e}{m_e|u|^2} - 1 \approx \frac{2T_e}{m_e|u|^2} \gg 1 \tag{64}$$

which contributes to the evolution of mirror modes through the growth rate (1) of ions as this growth rate contains the anisotropy of all electron components, including pairs, while the ion anisotropy $A \approx 0$ is about zero in the stable quasilinear state.

It is thus clear that a large electron pair anisotropy $A_{pair}$ will directly contribute to the growth of the ion mirror mode, once quasilinear stabilisation of the ion mode has eaten up the ion anisotropy, and the conditions in favour of trapping and pair formation have emerged. (In addition pair anisotropy will also destabilise the electron mirror mode in the stable quasilinear state of the ion mirror mode.) In both stable quasilinear cases the remaining ion (electron) anisotropy is negligibly small, and the quasilinear growth rate of the ion mirror mode is zero by definition.

With further evolution of the mirror mode starting from the quasilinear level, one must use the pair anisotropy in Eq. (1)

$$\gamma_{m,\,pair}(\boldsymbol{k}) \approx k_\| V_{A,ql} \sqrt{\frac{\beta_{\|i}}{\pi}} \left(\frac{T_{e\perp}}{T_{i\perp}}\right)^{\frac{1}{2}} A_e \tag{65}$$

including the expression for the electron anisotropy $A_e$. This applies to the quasilinear level where the depletion of the magnetic field is remarkable though not large, and the Alfvén speed $V_{A,ql}$ is based on the weak quasilinear magnetic field $B_{ql}$, while $\beta_{\|i}$ is on the quasi-linearly heated plasma level. Since, quasi-linearly, $T_{i\perp} \approx T_{i\|}$ has decreased to approach $T_{i\|}$, the temperature ratio $T_{e\perp}/T_{i\perp} \approx T_{e\perp}/T_{i\|}$ increases.

However, one has to determine $A_e$ as function of the pairs. Since $A_e$ is a pressure ratio, one must take into account also the non-paired isotropic electron component.

Let the fraction of pairs be $\alpha$, then one has approximately

$$A_e \approx \frac{(1-\alpha)T_e + 2\alpha T_e}{(1-\alpha)T_e + \alpha m_e|u|^2} - 1 \approx \frac{2\alpha}{1-\alpha} \tag{66}$$

which gives for the electron-pair generated ion-mirror growth rate

$$\gamma_{m,\,pair}(\boldsymbol{k}) \approx \frac{2\alpha\sqrt{1+\alpha}}{1-\alpha} \sqrt{\frac{\beta_{e,ql}}{\pi}}\, k_\| V_{A,ql} \tag{67}$$

where $\beta_{e,ql} = 2\mu_0 N T_e/B_{ql}^2$ is the electron-$\beta$ based on $B_{ql}$ at quasilinear stability, and the root $\sqrt{1+\alpha}$ arises from the combination of electron and pair pressures.

Since $\alpha < 1$, the pair induced electron anisotropy is not large but will be sufficient to break the quasilinear state and cause further growth of the ion mirror mode. This further growth violates pressure equilibrium and, if not immediately restored by breaking the pairs off and heating the plasma to restore isotropy, pressure equilibrium must be restored otherwise.

Under closed boundary conditions stabilisation proceeds again quasi-linearly through heating the plasma on the expense of the pair anisotropy $A_{pair}$ and possibly, in addition, by radiation of resonant whistlers. On the other hand, if open boundary condition allow inflow of cold charge-neutralised plasma from the environment, pressure balance will be achieved in this way by sucking in a selection of low-perpendicular energy particles of both signs. Mirror modes will in both cases quickly restore pressure balance, while growing to substantially larger than quasilinear amplitudes because, at any quasilinear level, new electron pairs will continuously be newly and readily produced.

## 5.5 Evolution of magnetic energy density

The magnetic energy density $W = B^2/2\mu_0$ will then about exponentially decrease

$$W(t-t_{ql}) = W_{ql}\left\{1 - \exp\left[2\gamma_{m,pair}\left(t-t_{ql}\right)\right]\right\} \tag{68}$$

from its quasilinear level $W_{ql}(t_{ql})$ until reaching its final minimum value $W_{min}(t_{fin})$ at final time $t_{fin}$ which is in equilibrium with the environmental pressure $P_{ext} = W_{ext} + \beta_{\perp,ext}$ according to

$$\frac{W_{min}}{W_{ext}} = \frac{1+\beta_{\perp\,ext}}{1+\beta_{\perp\,fin}} \tag{69}$$

Combining the last two equations gives an estimate for the time $t_{fin}$ when the mirror mode reaches its final equilibrium state

$$2\gamma_{m,pair}(t_{fin}-t_{ql}) = \log\left(1 - \frac{W_{min}}{W_{ql}}\frac{1+\beta_{\perp\,ext}}{1+\beta_{\perp\,fin}}\right) \tag{70}$$

With $t_{fin} \gg t_{ql}$ and $W_{min} \ll W_{ql}$ this expression simplifies to become

$$\gamma_{m,pair}t_{fin} \approx \frac{W_{min}}{2W_{ql}}\frac{1+\beta_{\perp\,ext}}{1+\beta_{\perp\,fin}} \tag{71}$$

Moreover, $\beta_{\perp\,fin} = (1+\alpha)\beta_{ql}W_{ql}/W_{min} \gg 1$. One then obtains the following estimate for the final saturation time

$$\begin{aligned}
\frac{t_{fin}}{\tau_A} &\approx \frac{\sqrt{\pi}}{2\alpha}\frac{1-\alpha}{(1+\alpha)^{3/2}}\frac{1+\beta_{ext}}{\beta_{ql}^{3/2}}\left(\frac{W_{min}}{W_{ql}}\right)^2 \\
&\approx \frac{\sqrt{\pi}}{2\alpha\beta_{ql}}\left(\frac{W_{min}}{W_{ql}}\right)^2
\end{aligned} \tag{72}$$

where $\tau_A = \left(k_\| V_{Aql}\right)^{-1}$ is the parallel Alfvén time based on the quasilinear saturation field $B_{ql}$.

In a mirror bubble of a few $10^3$ km parallel length, density a few $10\,\mathrm{cm}^{-3}$ and some $10$ nT internal field the Alfvén time

is of the order of $\tau_A \sim 1$ min, sufficiently long. Hence, if the final saturation time is al least about one Alfvén time we have for an estimate of the fraction of pairs in the mirror bubble

$$\alpha \approx \frac{\sqrt{\pi}}{2\beta_{ql}} \left(\frac{W_{min}}{W_{ql}}\right)^2 \sim \; \mathrm{O}(10^{-4}) \tag{73}$$

contributing both to the growth beyond the quasilinear saturation limit and maintenance of pressure balance with the external plasma and field, with measurable right-hand side.

This fraction is sufficiently small: Not more than roughly $\approx 10^3$ pairs per m$^3$ suffice for producing the desired effect, for instance in the magnetosheath, a rather small, unfortunately probably barely detectable fraction. Again similar to superconductivity, it is only the macroscopic effect of pairs that indicates their presence.

We have not included yet any probable inflow of plasma along the magnetic field, to help restoring the required pressure balance. Such an inflow is expected to occur, because the different mirror bubbles which form chains of bubbles along the magnetic field, again for instance in the magnetosheath, will evolve on different time scales. Being magnetically connected, they compete and tend to exchange low energy plasma along the field, which goes on their mutual expenses. Bubbles containing the largest number of pairs will grow fastest by sucking in plasma from the smaller ones in order to make up for their pressure balance with the environment.

An analogous mechanism should work also for the electron mirror mode which, on the quasilinear level, will grow fast, within few electron cyclotron times, and also attract cold electrons from the environment. This will all happen inside the ion mirror mode on the spatial electron scale.

In addition, the perpendicular pressure anisotropy of the pairs may in both cases, the ion as well as the electron mirror mode, excite resonant whistlers as well as Bernstein modes, the latter propagating in the perpendicular direction and having a characteristic banded structure following the electron cyclotron harmonics. Observation of these wave spectra, in particular Bernstein modes, should provide direct information on the presence of electron pairs. Since for pairs the fraction $q^*/m_e^* = e/m_e$ is unchanged, the electron cyclotron frequency is not affected. Those waves would, from the frequency, not be distinguishable from ordinary Bernstein or whistler modes, except for being restricted to the spatial volume of the mirror bubbles. It is clear that ion pairs, if formed, would behave similarly in the ion mirror mode. We have, however, not checked this possibility here.

### 5.6    Magnetic susceptibility

Above we developed a dynamical physical evolution model for the mirror mode, which allows it to grow beyond the quasilinear limit. The final state is an equilibrium that should be treated in the framework of thermodynamics or statistical mechanics of open systems. Open systems does not mean open boundary conditions. It means that the system is embedded into a very large system with that it exchanges information and possibly energy, and for that the distortion it causes is negligibly small, for example say, a mirror mode train in the magnetosheath.

In this ultimately achieved (observed or measured) equilibrium state the mirror mode should be described by thermodynamic quantities, and the localised expulsion of the magnetic field should be accounted for by a diamagnetic susceptibility $\chi$. This susceptibility

$$\chi_{pair} = \mu_0 \frac{\partial \mathcal{M}_{pair}}{\partial B} \tag{74}$$

is defined as the derivative of the total pair-magnetic moment with respect to the magnetic field, with the former not known and difficult to determine. It requires knowledge of the grand partition function $\mathcal{Q}$ of which it is the logarithmic derivative

$$\mathcal{M}_{pair} = T_e N_{pair} \left.\frac{\partial \log \mathcal{Q}}{\partial B}\right|_{N,T,\mu} \tag{75}$$

with $\mu$ chemical potential, which is related to the average density $\langle N \rangle$. Thus the susceptibility is the second derivative of the logarithm of the grand partition function with respect to the magnetic field at constant density, temperature, and chemical potential. Calculation requires knowledge of all energy states of the pairs in the volume and, in addition, the spatial pair distribution. The former can be restricted to only one state, the perpendicular final temperature $T_e$, while the spatial distribution requires model assumptions on the geometry of the ultimate mirror bubble and the diamagnetic surface current. Both are barely known. Thus we are basically unable to solve this fundamental physics problem, whose solution would provide insight into the most important thermodynamic connections in mirror modes, which clearly must hold under stationary conditions.

Referring to a heuristic approach, we can, however, at least determine the sign of the susceptibility when assuming that, to some extent, the total magnetic moment of all pairs that are localised in the mirror bubble would be proportional to the sum of all single moments of the pairs, while the dependence on the magnetic field reasonably remains unchanged. This may not be completely true though, because in analogy to superconductivity and superfluidity, the mirror mode diamagnetism is a second order phase transition and, therefore, the dependence on $B$ must be modified to some power $B^{-\delta}$, with $\delta \neq 1$ not an integer number. Neglecting this complication, we write

$$\mathcal{M}_{pair} = \alpha N \frac{\langle \mathcal{E}_{pair} \rangle}{B} \tag{76}$$

where $\langle \mathcal{E}_{pair} \rangle \sim T_e$ essentially is the electron temperature and, as above, $\alpha$ accounts for the fraction of pairs in the volume. Then we obtain that

$$\chi_{pair} = -\mu_0 \alpha N \frac{\langle \mathcal{E}_{pair} \rangle}{B^2} \tag{77}$$

This important result shows that the magnetic susceptibility is negative, which is required for observing a *local diamagnetic effect* in excess of the always present general, though weak and about unremarkable, bulk diamagnetism of any plasma that is caused by the orbital gyration of all the charged particles in the magnetic field and that is of the order of Landau diamagnetism (Huang, 1987), a fraction of Bohr's magneton.

Though the precise dependence on the magnetic field is not known, this argument suggests that the process leading to pair formation in mirror modes indeed causes the diamagnetism required for blowing up the mirror bubble, expelling the magnetic field, and causing chains of mirror bubbles.

Of course, such an estimate suffers from the involved uncertainties and the impossibility of constructing the conditions of phase transition in the high temperature mirror plasma which also accounts for pressure balance with the surrounding plasma and determines the final magnetic amplitude $B_{fin}$ in saturated thermodynamical equilibrium from the general relation $B_{fin} = B_{ql}(1 + \chi_{pair})$. This becomes an implicit third order equation

$$x^{\frac{3}{2}} \approx x - \alpha \frac{\beta_{ql}}{2} \tag{78}$$

for the final magnetic field, where $x = W_{fin}/W_{ql}$, and $\beta_{ql} = NT_e/W_{ql}$.

Neglecting the left-hand side yields approximately

$$\frac{W_{fin}}{W_{ql}} \approx \alpha \frac{\beta_{ql}}{2} \tag{79}$$

for a very rough estimate. From the previous subsection, which includes the condition of the required pressure balance with the surrounding environment, we conclude that the left-hand side is of same order as $\alpha$, say. Thus

$$\beta_{ql}/2 \sim O(1) \tag{80}$$

which, in spite of the severe assumptions made, is not an unreasonable value for the quasilinear equilibrium where pair formation sets on.

However, the main important result is the negative sign of the susceptibility which, independent on the real numerical value of $\chi(B)$, suggests that the mirror mode is not a simple plasma instability as such. Rather it is a particular plasma state occurring in high temperature plasmas. This kind of diamagnetism resembles a phase transition (Binney et al., 1999) whose precise physics has still to be developed.

Our discovery of the possibility of the involvement of electron pairing in the ion mirror mode in this case is an interesting though but a first step into that direction.

## 6 Conclusions

Mirror modes seem to be an exception in high-temperature collisionless plasmas. They start from a simple magnetohydrodynamic instability in an anisotropic pressure configuration far from thermal equilibrium that has been produced for instance in the magnetosheath (Constantinescu et al., 2003) by the forced flow across the bow shock (for a comprehensive review cf., e.g., Tsurutani et al., 2011; Balogh & Treumann, 2013, for the relations around the bow shock) and may be a general property of shocked plasma flows (Lucek et al., 2005). Linear theory shows that this instability produces magnetic field-elongated magnetic bottles which stabilise by quasilinear interaction between the anisotropic ions and the magnetic field in which course the thermal anisotropy is depleted and settles at a low stable rudimentary value. The amplitude of the magnetic depletion, as numerical simulations with periodic boundary conditions demonstrate, is very low. It is in fact so low that the quasilinear mirror mode would in observations not be noticed but added to the ordinary thermal fluctuations of the magnetic field and thermal pressure. It does not explain the notorious though not persistent observation of very large amplitude chains of mirror modes of up to 50% magnetic depletion.

Open boundary simulations (Shoji et al., 2012) show the evolution of large amplitude mirror modes with electron dynamics reduced to a neutralising fluid. Dynamics is purely ionic then, and pressure balance is provided by forced inflow of plasma from the external surrounding. This suggests that ions are capable of generating large amplitude structures, and that pressure balance is achieved by external plasma inflow. However, the assumption of fluid electrons is strong in this case and may not apply to real natural plasmas where electrons are far from being a fluid. For this to happen ion anisotropies must be large enough for pressure deficit to enforce inflow prior to quasilinear stabilisation. As noted above ion pairing would be a reason to generate such a deficit.

In the present communication we have not considered ion pairing except for pointing on its possibility. Instead we demonstrated that the physics of large amplitude mirror modes can be affected by a pairing mechanism which is unique in application to mirror modes as these provide the rare conditions where it may happen. We demonstrated this on the example of electrons but the theory can indeed be extended to include ions as well.

A possible resolution of the large mirror amplitude problem is thus found when accounting for the dynamics of the electron (or possibly ion) component trapped in the magnetic mirror bottle. Electrons perform their bounce motion and can, in the vicinity of their mirror points, where the parallel speed drops to near zero, get into resonance with the always present thermal ion-acoustic noise spectrum. Experiencing a modified dielectric constant they generate an attractive potential difference in their wake outside the charge-compensating Debye sphere that may affect another close electron, attract it and form an electron singlet pair which consists of the primary and attracted electrons in interaction with the resonant spectrum of ion sound background waves.

Formation of triplet pairs are of higher order and thus substantially less probable. The conditions for this to happen have been obtained. The electron pairs are not spin compensated because at the high temperatures in classical plasmas the Pauli principle plays no role. The remaining electron jitter energies are still high above spin energies but may be in many cases too low for letting the pair, which near its common mirror point $Z(s_m)$ is at average velocity $U(z_m) = c_s \ll v_e$, return into bounce motion as this would require that the jitter energy exceeds the attractive pair potential. We have not solved the stability problem of this process as this would become a separate investigation.

However, a group of pairs fulfils an important function in mirror modes. Trapped in attraction, they drop out of the bounce motion, become locked near mirror points along the quasi-linearly stable mirror bottle, and spend all their kinetic energy into their gyration. Thus the pair distribution in mirror modes becomes highly and narrowly peaked just above and near the perpendicular thermal velocity, an effect which is very interesting to investigate in all its further consequences. These may be manifold, since the pairs contribute a highly anisotropic population which, as noted, may become unstable to electrostatic and electromagnetic plasma modes. Among those are both whistlers, and electrostatic Bernstein modes. Whistlers are actually known to exist in observations and may be related to electron pairing. In the case of ion paring one would expect similar effects, in particular kinetic Alfvén waves.

Closer investigation has been given to two effect, the direct production of diamagnetism via the magnetic susceptibility (see the last subsection), and the contribution of the temperature anisotropy of the pairs to the envisaged further evolution of ion-mirror modes.

It has turned out that this effect may be realistic. It is based on two observations, the dependence of the ion mirror growth rate on the electron anisotropy which under normal conditions would not be of any interest as it provides just a small negligible electron contribution. However, in the case of quasilinear stabilisation the ion anisotropy is depleted and the growth rate vanishes. It is just this case when electron (and possibly ion pairs) are produced and come into play, as we have demonstrate above. (Considering the ion effect which we did not do leads to the result that ion pairs would restore an ion anisotropy and in this way break the quasilinear state and cause further growth of the mirror mode. This may have happened in the simulations of Shoji et al. (2012) where it neither has been discussed nor taken into account and where just large mirror modes have been demonstrated to develop in open boundary simulations.)

We suggested that electron pairs produce such an effect just in the ion mirror mode. This seems reasonable because of the high vulnerability of the mobile electrons to interact strongly with a background field. The production of even a small fraction of electron pairs breaks the quasilinear stability condition and causes further growth of the ion mirror in-stability. Pressure balance is restored as we have shown even for a small fraction of electron pairs per unit volume being created. It can also be warranted by forced inflow of cold plasma from the environment along the magnetic field, which probably happened in the cited simulations, and effect we are aware of but have not included in any detail here.

Altogether, the present communication discovered an interesting new effect in a high-temperature plasma which might have other consequences as well. It brings the theory of mirror modes to an intermediate physical conclusion by contributing to the so far badly understood generation of large magnetic amplitudes of mirror bubbles, the deep diamagnetic holes in the magnetic field. It also provides an important unexpected, at least interesting application of the apparently superficial attractive electron potentials in a plasma. Here we have demonstrated its possible importance in the evolution of mirror modes when electron-pair singlets form in close analogy to superconductivity.

It would be of interest investigating which effects the process elucidated here might have in turbulence theory and as well in astrophysical applications, in particular in view of our above finding that kinetic Alfvén waves would also be capable of generating attractive potentials to form electron pairs. Since they naturally have high phase speeds, satisfaction of the resonance condition seems to be natural for them with bouncing electrons for instance in the auroral magnetosphere. An example where kinetic Alfvén waves could become involved is auroral-plasma sheet coupling where electrons are naturally trapped in the geomagnetic field and perform large-scale bounce motions. The relaxed kinetic Alfvén pairing condition may generate a fraction of trapped pairs along the auroral magnetic field in this case to produce observable effects for instance in the aurora, generation of radiation, and in reconnection.

In mirror-modes kinetic Alfvén waves are of little interest, as there is no obvious reason for them to be generated. They have, moreover, never been identified in relation to mirror observations while ion sound waves are generally present within and outside them. In ion-inertial range turbulence kinetic Alfvén waves seem to play some role as various observations indicate and theory also supports for the reason that the scale of the ion-inertial range coincides with the perpendicular wave lengths of kinetic Alfvén waves. Our pair-singlet mechanism should work in those cases as well and might have consequences for turbulence, entropy generation, and turbulent dissipation.

*Acknowledgement.* This work was part of a brief Visiting Scientist Programme at the International Space Science Institute Bern. We acknowledge the interest of the ISSI directorate as well as the generous hospitality of the ISSI staff, in particular the assistance of the librarians Andrea Fischer and Irmela Schweitzer, and the Systems Administrator Saliba F. Saliba. RT also acknowledges the discussions on mirror modes with Dragos Constantinescu and Karl-Heinz

Glassmeier he had the fortune and pleasure to enjoy more than a decade ago at the Technical University of Braunschweig, Germany. Finally, we acknowledge the skeptical though constructive remarks of the referees.

*Author contribution.* All authors contributed equally to this paper.

*Data availability.* No data sets were used in this article.

*Competing interests.* The authors declare that they have no conflict of interest.

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
