# Peer review of "Electron pairing in mirror modes: Surpassing the quasilinear limit"

_Annales Geophysicae, 2019_

## Referee Comment (RC1) · Anonymous Referee #1 · 11 Aug 2019

In the manuscript, the authors construct a theory for the mirror mode as an application of the electron pairing mechanism at a spatial scale of around the Debye screening length in the presence of ion-acoustic waves. The theory predicts that the magnetic field within the mirror mode structure (or magnetic bottle) decreases at most by about 50%, which successfully explains the observations of the mirror mode structures in the Earth magnetosheath region.

This is an interesting manuscript, since the achievement in the manuscript can be summarized or oriented in two ways. The manuscript offers a nonlinear (large-amplitude) treatment of the mirror mode (motivating the plasma instability community and space plasma observation community); the manuscript demonstrates an application of the electron paring around the Debye scale in the spirit of the analogy to the celebrated

BCS theory for superconductivity.

The manuscript follows the earlier works by the authors themselves. The electron paring by over-screening in the presence of waves was studied in more detail by Treumann and Baumjohann (Ann. Geophys., 32, 975, 2014) by revisiting the pioneering works by Neufeld and Ritchie (1955) and Nambu and Akama (1985). Later on, the picture of mirror mode as an analogue to superconductivity is more clearly presented in Treumann and Baumjohann (Ann. Geophys., 36, 1015, 2018). The construction of the theory in the manuscript indeed reminds me of the BCS theory: paring the electrons (wake potential in sec. 3, two-electron potential in sec. 4), macroscopic effect by applying to the mirror mode structure (sec. 5).

The manuscript is worth for publication for various reasons. First, the manuscript deals with the Debye-scale physics, which is one of the frontiers in space plasma physics. Second, the manuscript advances our understanding of the mirror mode physics from the linear regime (small-amplitude) to the nonlinear regime (large-amplitude). Third, the manuscript bridges the gap between space plasma physics with the other branches of physics (in particular, superconductivity and phase transition). I raise some questions and comments in the following, and hope that the questions are solved before the publication.

Major comment

* I agree with most of the parts in the constructed theory. The major question I have is that if the paired electrons develop into a coherent motion in the manner of Bose-Einstein condensation (in the BCS theory) to produce a substantial amount of diamagnetic current and expel the magnetic field out of the magnetic bottle. I am under the impression, after reading section 5.2 and 5.3 many times, that the electron pairs can be incoherent from one pair to another, so not in the gyro-phase bunched fashion. Just a sufficiently large number of electron pairs must exist in the mirror mode structure. Could the authors comment on the coherence among the electron pairs?

[Figure]

Minor comments

* Another question I have is if the proposed theory is really meant for the mirror mode, or if the theory is also applicable to the magnetic holes observed, for example, in the solar wind. Magnetic holes or decreases differ from the mirror mode in that the structures are isolated and can appear in a solitary way in the magnetic holes, while the mirror mode is a plasma instability and is spatially periodic characterized by the wavenumber or wavevector (typically for the maximum growth rate).

* The number of the participating electrons (or electron pairs) to the diamagnetic current is estimated by the parameter alpha in the manuscript as N_pair = alpha N (line after Eq. 58 on page 19). I wonder if one could make use of the London penetration depth, lambda_L, (e.g., Kittel, Charles (2004). Introduction to Solid State Physics. John Wiley & Sons. pp. 273–278) to estimate or double-check the electron-pair number density from the magnetic field profile.

* How was Equation (7) derived? I understand the geometrical condition (equation at line 109), but the concept of temperature (or thermal energy because the Boltzmann constant is set to unity in the manuscript) comes abruptly in Eq. (7). I think some more explanation should be added as how the derivation of Eq. (7).

* Here are some comments about the language editing.

- page 2, line 58, "a combination of a classical plasma ion effect" with what?

- page 3, line 61, I am not a native English speaker, but I think "It are..." should read "It is ..." even though the essential subject is in plural, "electrons".

- page 3, line 78, "Instability..." should read "This instability..."

- page 6, line 170, "wave length" should read "wavelengths".

- page 6, line 173, "independent on" should read "independently of" or "independently from".

- page 7, line 203, "over-screening", I suggest to refer to Fig. 1 in Treumann and Baumjohann (2014) for the concept of over-screening outside the Debye sphere in the presence of acoustic waves.

- page 8, line 231, "at the contrary" should read "on the contrary".

- page 8, line 239, "consist" should read "consists".

- page 9, line 255, the question mark after "Bessel functions" should be resolved.

- page 9, line 264, "nominator", do the authors perhaps mean "numerator"?

- page 10, line 277, "sinus function" should read "sine function".

- page 10, line 282, "sinus function" should read "sine function".

- page 13, line 380, "at the contrary" should read "on the contrary".

- page 15, line 419, "an extended plasma" should read "a spatially extended plasma".

- page 15, line 427, "we need to now" should read "we need to know".

- page 15, line 428, "be tween" should read "between".

- page 18, line 511, "personal" should read "individual".

---

## Referee Comment (RC2) · Anonymous Referee #2 · 2 Sep 2019

The manuscript discusses a possible mechanism to support formation of large amplitude mirror mode or magnetic bottle elongated along the magnetic field. Generation of electron pairs consisting of two electrons is assumed, and necessary conditions for their existence are discussed based on electrostatic potentials and interaction with ion sound waves. An electron pair is assumed to from a coherent gyration current at which reduces the magnetic field inside the mirror bottle. The idea sounds interesting, but it is not convincing to assume the formation of the electron pairs based on the theoretical explanation presented in the manuscript as commented below.

1. First of all, it is not clear why we need to study electron dynamics for the mirror mode which can be supported by ion dynamics only. Temperature anisotropy of thermal ions and its thermalization in the nonlinear stage may well be good

[Figure]

enough to support the large amplitude mirror modes in the magnetosheath. For example, a hybrid code simulation [Shoji et al., 2012], where electrons are treated as a fluid, shows formation of large amplitude mirror mode waves. These mirror mode structures are very dynamic in the time scale of hundreds of the ion cyclotron period repeating nonlinear coalescence of magnetic structures initially formed by the linear mirror-mode instability driven by temperature anisotropy of the thermal ions. In such evolution of the mirror mode magnetic fields controlled by ion dynamics, assumption of formation of electron pairs in an ultimate state of quasi-linear equilibrium is not realistic.

2. In considering electron pair formation, the authors only assume electrostatic forces and gyro-averaged mirror force. However, there should be some Lorentz force due to the perpendicular velocities and transverse wave magnetic fields such as whistler mode waves. In addition to resonances with the ion sound waves, resonances with other eigenmodes in the magnetic bottle (whistler mode wave and Langmuir waves) should be considered.

3. Description of the coherent gyration current $J_{pair}$ is insufficient and difficult to understand. What is the relation of gyro-phases of the two electrons forming the pair? Can they be stable? Generally, a nongyrotropic distribution of electrons is strongly unstable generating electromagnetic waves from the transverse current.

4. Analogy with the Meissner effect in superconducting metal is used all through the discussion. More quantitative confirmation by simulations is necessary to make the readers convinced with the proposed mirror mode physics.

Reference
M. Shoji Y. Omura, and L.-C. Lee, Multidimensional nonlinear mirror-mode structures in the Earth's magnetosheath, J. Geophys. Res., 117, doi:10.1029/2011JA017420, 2012.

---

## Referee Comment (RC3) · Dragos Constantinescu (Referee) · 3 Sep 2019

The manuscript investigates the influence of electron pair formation on the evolution of linearly saturated ion mirror mode. The Authors derive the diamagnetic effect of trapped electron pairs which is proposed to drive the growing of the mirror mode amplitude from a low perturbation in the magnetic field, reached when the initial anisotropy is depleted, to the observed levels comparable with the ambient magnetic field magnitude. By providing a theoretical basis for the observed large amplitudes of the mirror mode, the present work represents a significant contribution to space plasma physics.

The mechanism of electron pair formation proposed in the manuscript is based on the attractive potential due to overcompensation of the charge occurring just outside the

Debye sphere of a moving electron in interaction with a parallel propagating ion sound wave. It is shown that in order for a negative potential to develop, the parallel component of the electron velocity must be close to ion sound wave phase speed. Since the thermal speed of electrons in space plasmas is much larger then the ion sound wave phase speed, this effect is in general of little importance. However, the Authors argue that special conditions encountered inside the quasi linearly saturated magnetic bottles favour the resonance of the trapped electron population with the always present background of ion sound waves. As trapped electrons approach their mirror points, their parallel velocities decrease until the resonance condition is satisfied and a negative potential develops. Another electron moving at almost the same velocity can be captured in the potential well and form a pair.

Minor points

A key assumption throughout the manuscript is the conservation of the magnetic moment of the electrons. However, as the magnetic bubble grows and the field decreases, the gyration radius of the electrons in the low field regions increases. It is known that a necessary condition for the magnetic moment to remain invariant is that the gyration radius of the particle must be much smaller then the curvature radius of the magnetic field. If this condition is not satisfied, the motion of the electron changes from a periodic bounce between the mirror points to an irregular motion. I think this should be mentioned in the paper, or even better an estimate of the fraction (which might be significant) of particles which become irregular and therefore do not participate in the pair forming should be given.

page 4 lines 107 - 114

Electrons with energies larger than $\mathcal{E}_{trap}$ will indeed escape from the bottle *orthogonal* to the magnetic field, however, not all electrons with energies bellow $\mathcal{E}_{trap}$ will be trapped, they might escape of course along the magnetic field if the pitch angle is too

small ($|s_m| > \pi/k_\parallel$). This will affect equation (8) on page 5.

page 18 lines 505-516

The authors should explain the reason for the newly form pair to "drop out" of the bouncing motion and remain locked near the mirror point. The center of mass velocity $U$ must be slightly bellow the ion sound speed for the electron pair to form. If the bouncing motion continues then this condition is quickly broken and the pair disintegrates. If the pair is locked near the mirror point, $U$ vanishes and again the pairing condition is broken. Please clarify how the electron pairs can remain stable, or is rather a dynamic equilibrium at work, where disintegrating pairs are compensated by newly formed ones?

page 18 lines 530-555

It would be useful to discuss how does the diamagnetic effect of the electron pairs compare with the diamagnetic effect of the former "free" electrons before pairing

page 19

Equation (59): after the first "=" sign there should be no "−" sign

---

## Referee Comment (RC4) · Anonymous Referee #4 · 6 Sep 2019

This is a very interesting paper presenting an unorthodox view on the nonlinear state of the mirror instability. The authors build on the previous work of studying the attractive potential of electrons in plasma due to interaction with plasma waves. They propose that under special conditions, when in resonance with ion acoustic waves, the Debye shielding of a point charge can result in a positive effective potential and thus attract other electrons. In an analogy with the theory of superconductivity, the authors propose that the electrons can form gyrating pairs which in turn exhibit a diamagnetic effect due to their magnetic moment. This effect is proposed to reduce the magnetic field inside mirror structure, being responsible for the observed shape and amplitude of mirror modes.

While I think that the paper is an interesting contribution to nonlinear mirror mode

physics and as such is definitely worth publishing, there are some crucial points which are not clear to me and perhaps should be addressed in a revision.

1) Perhaps I do not understand the reasoning properly, but from section 5.1 it seems that the centers of mass of the electrons continue in their bouncing inside the magnetic mirror. On the other hand, on lines 505-507, it is said that the pairs drop out of the bouncing motion and are locked in their position. This is not clear to me. How does it follow from section 5.1 ? Could you pleaswe clarify ?

2) While I understand how the electron pair can be formed, it is unclear how they can be sustained for a longer time. The attractive potential can only be present while in resonance with the wave. Bouncing electron pairs would get out of resonance as soon as they move away from the vicinity of point s_m. Assuming that the electrons are indeed locked in the appropriate location (see point 1 above), doesn't maintaining the attractive potential require some pre-existing ion-sound wave which the electrons are in phase with ? I would expect the phases of the thermal ion sound waves to fluctuate and get out of phase quickly.

Is the jittering motion of the electron pair compatible with being in resonance with the waves ?

3) My main comment is to the discussion of magnetic susceptibility in section 5.3. I do not understand how the magnetic moments of the gyrating electron pairs differ from the magnetic moments of ordinary trapped gyrating electrons. The gyrating pairs certainly contribute some magnetic moment, but so do single gyrating electrons with a similar perpendicular velocity. The authors make references to coherent motion, but little justification or explanation is provided, apart from vague a reference to the theory of superconductivity.

Minor points:

- line 61: "It are ..." sounds wrong - line 450: there is probably a word missing after

"magnetic". The sentence is thus unclear. - equation 34: the variable y should be defined.

---

## Author Comment (AC1) · 6 Sep 2019

General Comment:

1. We thank all four Reviewers for their efforts in evaluating our paper and their (mostly) constructive comments.

From reading the reviews it becomes clear that this whole item is fairly difficult and by far not ultimately explored. We are very well aware of this but have chosen to nevertheless inject the idea into the space plasma community for its unexpected fundamental physical interest and probably also its singularity in application.

We have chosen to formulate this general comment as an answer to the reviewers for making it easier for them to read (and for us to write) the general (anyway overlapping)

response, which is detailed enough, than a response to each minor or larger point in the 4 reviews. The reviewers essentially raise all the same questions. So it makes sense to write a common response.

Nevertheless we also provide brief responses to each reviewer just focussing on their main points. The demanded minor corrections we will do without noting them explicitly. We believe that in science one should have sufficient trust in each other such that those which we consider worth correcting/changing will have be done by resubmission.

In the MS we have made a number of changes (according to the suggestions of the Reviewers and in the spirit of these general comments). All changes are in blue for easy reading.

2. It indeed seems that mirror modes are a rare case, possibly the sole case in high temperature plasma physics where an effect like the one proposed here may take place: pair formation resembling a quantum effect known from superconductivity but this time in a classical system. The idea is based on early work in the 1950ies referring to plasma oscillations (Langmuir waves) which was intended to apply to superconductivity but did not work. It was formally picked up by one later Nobel laureate, Nambu, to show that instead ion sound (exactly the phonos in superconductivity) instead of Langmuir waves would be more promising. But there was nowhere any application in sight.

We have made these calculation more precise to derive the exact and applicable conditions for pairing. We also have shown that the pairing condition can be applied formally to any other plasma wave, even electromagnetic waves. Though this is interesting, it does not mean that it would work. In fact it will not for most waves. The basic problem with this entire approach is the required resonance condition. For electrostatic waves it relies on the approximate equality of the wave phase velocity with the particle speed, a condition very difficult to satisfy because in high temperture plasmas electrostatic phase velocities are generally small, and thus the fast particles will barely interact or
enter the required resonance. Electromagnetic waves have higher phase speeds but for them the electric field is perpendicular to the propagation. The required resonance supposes gyrobunching and thus will be rather ineffective. In any case conditions for attractive potentials will be sparse. One of the reviewers complained the restriction to electrostatic waves. His complaint safely be rejected. The exception is KAW which we did not investigate.

3. However, considering the bounce motion of electrons (or ions) in mirrors modes, once they have developed and are of sufficiently large amplitude, which we assume is the quasilinear stable state, is a promising case, possibly the only promising case in plasma. Bouncing particles (electrons and as well ions) have sufficiently low parallel speeds around their mirror points for going into resonance with slow parallel propagating electrostatic waves. This applies only to their parallel speed. This makes attractive potentials probable for a number of the mirror trapped bouncing particles. These attractive potentials are useless of they do not attract another particle of same charge.

However, for attracting another particle one needs to include its effect. Therefore our efforts to calculate the two particle case (the singlet) which resembles Cooper pairs. Three electron interactions (triplets) are improbable. We have checked this but not included. The triplet probability is low, this means, the trapping condition is highly restricted to a narrow range of parameters only, so not being of any importance. We also restricted to electrons here for obvious reasons. The claim to include ions (as put forward by one of the reviewers again) is not unreasonable but not necessary at this stage. For ions similar conditions should hold which are modified by the ion mass and cause different potentials. Calculation would go along same lines.

4. Staying with electrons it becomes clear from our calculation that two electrons with close-by mirror points and in approximate resonance with ion sound (in our case because of the parallel electric field and phase speed) can under the derived conditions form pairs. For any equatorial electron pitch angle distribution inside the mirror there will be a distribution of bounce-mirror points along the magnetic field, and hence a

distribution of pairs along the magnetic field. Calculating it requires inclusion of the pitch angle distribution (which we did not do at this stage of the theory). Once the pair has formed it has nearly same speed as the ion sound plus a small oscillation around the mirror point. If the pair is stable, which is quite a different question, which we did not investigate but which is of importance, requiring further analysis, then all energy of motion is in the perpendicular speed with very little left for the oscillation around the mirror point.

We have shown, that the electrons which form pairs must have energies above but very close to thermal energy. Hence their perpendicular speed is roughly thermal, from which follows that they have all about same gyroradius, depending only on the location in the magnetic field. In this sense they are coherent while not bunched by no means, as the phase of the gyration plays no role for the pair. It is their parallel motion which determines the paring.

5. The question is whether they can escape or not from the mirror points in order to return into bounce, a question raised by two reviewers. This is the question of stability. Since this requires solving the stability problem, a different paper, we have not attacked it. However, there is an interesting argument: at the mirror point $Z=s\_m$ and if they remain in resonance with the ion sound whose velocity is unaffected in this interaction, then they will become locked at the mirror points and drop out from bouncing. Locking depends on the energy in the rudimentary oscillation velocity u around the mirror point. The pair here at its common mirror point has zero parallel speed U=0. So it needs to become catapulted out in order to continue bouncing. If the remaining energy in the jitter motion $u^2$, the small amplitude oscillation, exceeds the trapping potential, then the mirror force will take over and the trapping will not be stable, the restoring force of the bounce will destroy the pair and reinject the two electrons into the general bouncing. For some of the electronic population this will be the case. However, for some it will not, becasue being in resonance with the ion sound phonon (which has nothing in common with the bounce force) these pairs with small $u^2$ become extracted

from bounce and will be stable and consequently become locked to the phonon for comparably long time near the mirror points s_m.

In addition, of course, it is also possible that the destroyed pairs will be replaced by new ones which form in interaction with other ion sound waves, the equivalent in Cooper pairs which form and decay and form anew. Since the particles have no identity this replacement implies that there will always be a certain number of pairs even though this number may fluctuate. Thus one expects that a continuous population of (electron) pairs exists in a mirror mode. Either electrons, which we investigated, or ions which also may form pairs under modified conditions which we dared to investigate here.

6. Now there are a number of questions:

First, is it necessary to include the Lorentz force (as one of the reviewers demanded)? The answer is clearly no. For pair formation it is definitely not necessary as long as one does not refer to electromagnetic waves (see above) and stay in the nonrelativistic domain (which we did). So at this stage it seems unnecessary to call for the Lorentz force. That part of it which refers to the bounce, the mirror force, is incuded in the calculation by reference to the bounce motion. The above arguments suffice in this respect.

Second: we took into account ion sound waves for two reasons. They are almost always present and have parallel relatively high phase speeds in favour of application in mirror modes to electrons. Most other electrostatic waves have perpendicular electric fields, Bernstein modes for instance. Electromagnetic waves, whistlers for instance have favourably high parallel speeds but perpendicular electric fields as well. Hence their interaction with electrons is strongly modified and much more complicated. Resonance requires cyclotron resonance and thus is restricted to a different group of electrons in spite of their advantage, the higher phase speed. In principle the basic equation for the potential holds also for them as well (we have given the condition in the paper) but the resonance condition must be reinterpreted. We do not see any reason to do this at this stage. This would be a totally different investigation. The same argument applies for ions to ion-cyclotron waves.

Third: we have not investigated the stability of electron pairs but in the above gave arguments that at least a group of electrons will build up stable pairs.

Fourth: we have not considered ions so far. Their interaction with ion sound might possibly cause similar effects. The potential will be different though because of their mass, but they favourably have slower velocities which in pairing might make them lesser dependent on bouncing than electrons which, however, is not known to us whether it is favourable or not. Possibly their contribution is important in view of the application to mirror modes where they are anyway trapped and bounce as well around. If they pair, however, their reduced deendence or even independence on the mirror points will not have a positive effect on the evolution of the mirror mode. This contrasts electrons where some pairs remain locked at the mirror points. But at this stage we refrain from applying the theory to ions even though it seems simple. The formal equation will be the same. We are happy with having solved the difficult problem for singlet paired electrons.

Focussing on electrons was central to our approach. However the basic equations hold as well for ions and for any other plasma wave which satisfies the resonance. This is quite clear physics. Hence, the theory, with ittle changes, applies to them. We have not checked the effect of their larger inertia. Both possibilities exists that ion pairs are less or more stable. In any case their effect might be similar whether stronger or weaker is a question of a separate investigation.

Fifth: We have not investigated in any detail the role the large perpendicular anisotropy of the pairs plays in the evolution of other instabilities, in particular the electron mirror mode. We have just mentioned the fact in the paper.

It is however clear that if a large number of electron pairs is locked in gyration, then the large anisotropy will contribute mainly to the electron mirror mode which is smaller

scale and has been observed. Maybe its observation at comparably large amplitude is already proof of the reality of electron pairing. This we have not investigated. In addition the electron anisotropy could also exite resonant whistlers, which have also been observed.

Sixth: Our application to the amplitude of mirror modes is just heuristically motivated. It is based on the assumption that a sufficiently large number of electron pairs is locked in gyration with nearly same gyration speed which implies a current. We were speaking about "coherence" but do not mean phase coherence, just gyration coherence. The pairs are not bunched but contribute to a surface current whose magnetic effect is diamagnetic thus increasing the depletion of the magnetic field. This we have heuristically accounted by estimating the susceptibility. It is clear that pressure balance in this case must be provided by instreaming of untrapped plasma along the magnetic field from outside the mirror bubble. We simply assumed that pressure balance is given. However this theory is preliminary.

Seventh: there is the question on what we called coherence. It should be clear to everyone that coherence here is not phase coherence or bunching. Locking in pairs and escaping from bounce by having all energy in the perpendicular at almost same thermal speed means that all pairs gyrate at about same gyroradius for the field value at a given $s\_m$ forming a shell of nearly common radius. This shell thus carries the diamagnetic surface current whose effect we estimated giving a heuristic argument. This is meant by coherence, not phase coherence which is nonsensical as it is not generated by pairing.

Eighth: The final question we address here (leaving aside a number of others) concerns pressure balance. Clearly quasilinear theory provides pressure balance. However if additional expansion of the magnetic field is generated then pressure balance must be restored. A minor contribution is caused by magnetic stresses in the surface current, as is well known. However the main effect is due to inflow of quasi-neutral plasma from the environment along the magnetic field. This plasma has very small

perpendicular energy, less than required for bouncing. Magnetic moment conservation will cool it in perpendicular direction when approaching the centre. However, its mere number increases the pressure in the mirror bottle to account for pressure balance. So, pressure balance will be warranted in this way.

It would be very interesting if observations could be designed to check these proposals or to vindicate them. If it could be confirmed experimentally that pairing in a classical plasma is possible as proposed here, then mirror modes would be the ideal place to check for all the effects listed. Maybe observation of large amplitude electron mirror modes and localized large amplitude whistlers, both excited by the large perpendicular anisotropy of the pair component, are indications of pairs. In any case, identification of pairs is an interesting task to be performed experimentally because of its deep physical meaning.

Once more, we express our thanks to the intriguing and important comments of all 4 reviewers. We have answered here the main questions which concerned mostly around the non-solved problem of stability of pairs. Our investigation is a first step in that direction, while investigation of stability would be the next step before including the pitchangle distribution, constructing the real spatial distribution of pairs and for the stable pair population developing the microscopic theory of the magnetic susceptibility. We shall includ changes accordingly (in blue again) into the resubmission.

---

## Author Comment (AC2) · 7 Sep 2019

We thank the reivewer for the favourable comments.

We would like to direct him to the General Response we have posted in the Discussion where most questions he has have been answered in the interest of all 4 Reviewers. Here some brief specific remarks on the questions raied by the report:

1. It is absolutely true: the pairs are not phase bunched. They are just all in gyration only which is sufficient for the surface current. Phase bunching cannot be achieved and is not necessary. It suffices that at zero parallel speed the surviving pairs are on a ccommon shell. this we mean by coherence. We have said this now explicitly in the paper and weakend the expression o quasi-coherent.

[Figure]

2. The mirror mode provids the ideal conditions for pair formation. We would believe that this is a rare case because of the bounce motion. However, it might be that in magnetic holes which also probably permit for bouncing ans as well other places, say reconnection geometries or also the auroral magnetosphere (i.e. in all cases of magnetic trapping or quasi-trapping like in reconnection where the particles bounce between plasmoides) the possibility exists for this kind of pairing as it only requires reflectin at mirror points with conseration of the magnetic moment, presence of waves to resonate (here ion sound, otherwise parallel propagating waves mostly of electrostatic nature. The formal expression for the potential allows for all kinds of waves but the resonance condition sets another requirement which is not easy to satisfy.

3. Yes, of course the London length could be used. However, at the present state of the theory which in these last subsections is rather speculative as it just suggest a possiblity, I would dare to apply it. The factor $\alpha$ is purely heuristic and very uncertain yet. Before doing it one would have to solve the stability problem of the pairs which is a difficult task which we did not attack at this moment. At this time the value of $\alpha$ is rather large, and one would expect that it is much less because some pairs will dissolve, being replaced by new pairs such that the number will strongly fluctuate. The London length itself is very uncertain, just a rough parmeter even in superconductovity theory which is certainly more precise than plasma theory at the high temperatures where for instance pressure balance is not better than roughly 10

4. Eq 7, thanks this is trivial. Slightly changed.

5. Thanks for the language corrections.

---

## Author Comment (AC3) · 7 Sep 2019

The "Response" posted was meant to Rev 1.

Unfortunately it was incomplete when copying it in an doing Latex which erased everything after the % sign.

Here is the missed part:

better than 10% or so. The value of pairing is therefore less in the speculation on the susceptibility made in the last subsections than in the contribution of a perpendicular temperature/pressure anisotropy. This should drive other instabilities of various kind if possible in the mirror plasma. Those are in the first place electron mirror mode which have ultimately been observed again at much larger amplitude than quasilinear, but

also other waves Bernstein modes, whistlers (which have also been observed) etc. Maybe observation of these modes already proves the existence of electron pairs? It has not been checked ut should. Also in application to ions one could put forward similar arguments. The interesting point is in any case that once the quasilinear limit is reached which is in pressure balance as it simply heats the trapped component on the expense of the perpendicular anisotropy (whether electrons or ions) until pressure balance is achieved, the additional anisotropy of the pairs (electrons or ions) causes a pressure imbalanced depletion of the magnetic field which must be pressure compensated which happens mainly by sucking in additional cold nonbouncing plasma of small magnetic moment from the surrounding. If this is forbidden, then one would expect that additional heating would be produced, and this would eat up the perpendicular anisotropy caused by the pairs, putting them back into the plasma and restoring pressure balance. In this case existence of pairs would be an intermediate state which leads to large mirror amplitudes (applicable to both electrons and ions).

---

## Author Comment (AC4) · 8 Sep 2019

Thanks very much for the important comments.

We would like to direct you to the General Response we made were the overlapping questions of all 4 Reviewers have been discussed.

Here just some specific responses to your points which are all very reasonable.

1. 2. We believe that in the General Response we have given arguments to this point. It is a question of the stability of pairing (which is implicit in your criticism) which we did not attack at this point just staying with a simple argument. In principle, if no pairing would exist the two electrons would simply continue their bounce motion after having reached their common mirror point $Z = s_m$ being reflected by the mirror force. At $s_m$

they have as a pair zero parallel speed, just the small jitter velocity. Hence if the energy in the jitter is less than the trapping potential the pair will be stable and cannot return into bounce because no force acts on it to drive it back. This is the stability problem said in a few words. Thus for some stable pairs there is the possiblity to become locked at the mirror point, others will return to bouncing, maybe after some rest, maybe immmediately, some will dissolve. However, always new pairs will form such that in the average there will be a stable pair component present. This is the same in BCS of superconductivity where the pairs come and go but in the average exist as a population. However, the proof of this point requires solving the stability problem which we did not do yet as it is not just an easy task. Locking itself is very interesting, and we believe that mirror modes are a rare case where this kind of physics is at least imaginable and possible as kind of a very rare macro-quantum state in high temperature plasma physics where such effects are not expected normally.

Concerning the ion sound spectrum: there is a broad spectrum of ion sound for instance in the magnetosheath (the place where most mirror modes have been observed) as background noise. Any pair will always find a group of waves with that it can resonate. We have taken this into account by integrating over the whole $k$ spectrum of ion sound waves in the potential expression. So this is not a big problem in fact. In looking at obervations (cf. Rodriguez and Gurnett 1975 or ours Treumann and Baumjohann 2018 on the electron mirror mode) the ion sound spectrum is indeed about constantly there and is sufficiently broadband to affect some part of the bouncing electron (or also ion) distribution (the latter we did not consider, however the equations are formally the same with exception of different speeds and inertial effects).

The question on the jitter motion we have not attacked. We assumed that $u \ll V$. However, the range $u$ determines the range of $k$ which is in resonance. Again, when $u$ becomes large this range may not exist, and the pair will dissolve rapidly because the potential will decay. These are of course all questions which cannot be answered easily nor calculated easily. Our assumption is that $u$ is sufficiently small. As in all

those theories there is always some uncertainty in the assumptions.

3. Don't put too much weight on the susceptibility. This whole part is highly speculative. However to explain what we suggested:

Of course, any plasma containing gyrating charges has some diamagnetism which is however very weak and distributed over the entire volume such that it is not remarkable (the famous incoherent Landau diamagnetisms per particle which is just one third of the Bohr magneton).

Our argument is that the pair current is restricted to a shell of some comparably narrow width $\Delta r$ because all pairs have the same orbital speed $v_e$ (the gyration phase plays no role because at both bounce and mirror frequencies it is unimportant). Our term coherent is meant in just this restricted sense.

The concentration of pairs in this shell implies a surface current whose magnetic effect should locally become measurable. Still the effect might be weak and probalby is, because the fraction of pairs $\alpha$ is probably much less than estimated on the assumption that its yield would be 50% of the mirror mode. The idea is rather that this probably as well small diamagnetism might start inflow of plasma from the surrounding such that the quasilinear threshold can be overcome. The final state is then not determined in the way we proposed but differently by how much plasma of low perpendicular energy flows in from outside.

Actually, at this point in application to mirror modes, one should turn to include ions as our theory would directly apply only to electrons and thus to the electron mirror mode. The equations however also hold for ions though at different time scale. We did not dare to apply them to ions but this is possible (and demanded even by one of the reviewers). Similar effects may be expected. With ions definitely the magnetisation would be completely unimportant as only electrons play a role in the magnetisation susceptibiity.

[Figure]

The physics would then not go the way on calculation of the magnetic susceptibility but on the inclusion of the additional large perpendicular anisotropy provided by the pairs (both electrons and ions) which restarts the mirror instability at the quasilinear level to continue growing and exceeding the quasilinear limit. This causes an uncompensated pressure deficiency in the mirror mode which needs to be compensated by sucking in low perpendicular energy particles (quasi-neutral ions and electrons) along the field from the outside. It seems that this would be a more realistic mechanism.

We shall rewrite this part of the paper in this spirit in order to prevent misunderstanding and over-valueing the magnetisation. Though thermodynamically this would be much more interesting and important as a physical effect and was reason for our excitement, it is probably unrealistic. On the state of the physics of our days there are very little effects left which can cause real basic physics excitation.

The pure fact of the realistic possiblity of pairing (singlet states) in mirror modes in high temperature plasma should be sufficient but should not lead to exaggerations in expectations.

Once more many thanks for your comments which we hope have been clarified by the above responses at least to our own satisfaction.

---

## Author Comment (AC5) · 8 Sep 2019

Many thanks for the excellent report, Dragos.

First, we would like to direct you to the long General Comments to the overlapping comments of the Reviewers where most of the questions is answered.

However your particular questions we will take into account in resubmission.

Yes, indeed, mirror modes seem to be the rare case where an effect like this can, in principle, become realised. Not only for electronds but under modified conditions also for ions (which we didn't dare to include here but what could be easily done by reformulation and might even be better in application to ion mirror modes while electrons possibly apply better to electron mirrors which, interstingly enough, have also been

observed at too large amplitudes scattered over the ion mirror modes —- it would be interesting to check conditions, where only electron and no ion mirrors develop, i.e. conditions of isotropic ions and anisotropic electrons). Ture, the bouncing is the key ingredient by reducing the parallel speed which is needed for resonance with the waves. It is clear that one needs electrostatic waves, alle electromagnetic have perpendicular electric fields and thus are not suited for attractive potentials. But the key observation is that one needs to do the calculation for two electrons, in order to get pairs.

The other problem is locking. We gave the main arguments in the General Response why locking occurs: it is, as you correctly noticed, a question of stability. We have explicated the reason in the General Response. Briefly: locking at $Z = s_m$ for the pair occurs for two reasons: first their we have $U \approx c_s \approx 0$ thus only $u \neq 0$ in the jitter motion (which, rightly as you say, is irregular). Returning into bouncing requires that $mu^2 > \Phi$, i.e. the energy in the jitter must exceed the trapping potential. If this happens then the mirror force can katapult the pair out of resonace and let it return into bounce. If not the pair will, at least for some time, be locked. Thus there will be a number of pairs which decay, another number which is stable for some time. However, sinde the particles have no identity always new pairs will form and others decay, so one will have s fluctuating population of pairs present all time. This is the main point.

The other reason for being locked is that the ion sound does not participate in bouncing. Thus once the three particles (2 electrons plus ion sound) interact and a sufficiently large trapping potential evolves, the ion sound wave does not easily allow the pair to return into bounce becasue this would require turning the waves around. But all of this is a question of stability analysis. Some pairs will decay, some will as pairs return to bounce (those with large $u$) and some will be stably loked. It is not that $U$ vanishes completely but that $U = c_s$ becomes close to the mirror point (just before $U = 0$) where the pairing occurs. This means that the pair is locked to the slow ion sound (nearly zero velocity), negligible with respect to the thermal speed in the perpendicular motion.

The question on the diamagnetic effect is intriguing. We were probalby overoptimistic.

In any magnetised plasma there is a global diamagnetic effect slightly reducing the magnetisation. But this effect (Landau diamagnetism in a homogeneous plasma) is not remarkable because it is global distributed over the entire domain and in weakly inhomogeneous plasmas also not remarkable. Our expectation was that the restriction of the motion to a current shell by locking would concentrate the diamagnetic effec locally. This might be, as I said, too opitimistic. So the heuristic section on the magnetisation is not relevant. Since it is very hard to make the relevant calculation in the inhomogeneous case (surface current plus its stresses produced by an unknown distribution of pairs), the heuristic estimate of the factor $\alpha$ is probably too large, possibly way too large.

The realistic idea is probably that the large addition perpendicular temperature anisotropy introduced by the generation of locked pairs at the quasilinear saturation level, which is in pressure balance with the heated population, destroys the quasilinear stability and drives the instability towards larger magnetic depletion. There are two possibities. Either this causes additional heating and new quasilinear saturation at higher depletion while by erasing the pair additional anisotropy also destroying the pairs (new pairs may form on the hotter plasma background such that this process could in principle continue), or colder (at perpendicular speeds below trapping/reflection/bounce) neutral plasma is attracted (sucked in) along the magnetic field and contributes by hsear number to enhanced pressure to come up for pressure balance.

These are two realistic possibilities, both causing deeper mirrors and both based on pair formation. This is most interesting in both (for electrons and ions as well) cases which would be open to experimental investigation: identification of the sucked in flow and the pair populations.

Thanks also for the minor comments.

---

## Editor Comment (EC1) · Nick Sergis (Editor) · 11 Sep 2019

We thank the reviewer for the favorable comments. We would like to direct him to the General Response we have posted in the Discussion where most questions he has have been answered in the interest of all 4 Reviewers. Here some brief specific remarks on the questions raised by the report:

1. It is absolutely true: the pairs are not phase bunched. They are just all in gyration only which is sufficient for the surface current. Phase bunching cannot be achieved and is not necessary. It suffices that at zero parallel speed the surviving pairs are on a common shell. This we mean by coherence. We have said this now explicitly in the paper and weakened the expression o quasi-coherent.

[Figure]

2. The mirror mode provides the ideal conditions for pair formation. We would believe that this is a rare case because of the bounce motion. However, it might be that in magnetic holes which also probably permit for bouncing as well other places, say re-connection geometries or also the auroral magnetosphere (i.e. in all cases of magnetic trapping or quasi-trapping like in reconnection where the particles bounce between plasmoides) the possibility exists for this kind of pairing as it only requires reflecting at mirror points with conservation of the magnetic moment, presence of waves to resonate (here ion sound, otherwise parallel propagating waves mostly of electrostatic nature. The formal expression for the potential allows for all kinds of waves but the resonance condition sets another requirement which is not easy to satisfy.3. Yes, of course the London length could be used. However, at the present state of the theory which in these last subsections is rather speculative as it just suggest a possibility, I would dare to apply it. The factor $\alpha$ is purely heuristic and very uncertain yet. Before doing it one would have to solve the stability problem of the pairs which is a difficult task which we did not attack at this moment. At this time the value of$\alpha$is rather large, and one would expect that it is much less because some pairs will dissolve, being replaced by new pairs such that the number will strongly fluctuate. The London length itself is very uncertain, just a rough parameter even in super-conductivity theory which is certainly more precise than plasma theory at the high temperatures where for instance pressure balance is not better than better than 10% or so. The value of pairing is therefore less in the speculation on the susceptibility made in the last subsections than in the con-tribution of a perpendicular temperature/pressure anisotropy. This should drive other instabilities of various kind if possible in the mirror plasma. Those are in the first place electron mirror mode which have ultimately been observed again at much larger am-plitude than quasi-linear, but also other waves Bernstein modes, whistlers (which have also been observed) etc. observation of these modes already proves the existence of electron pairs? It has not been checked it should. Also in application to ions one could put forward similar arguments. The interesting point is in any case that once the quasi-linear limits reached which is in pressure balance as it simply heats the trapped

component on the expense of the perpendicular anisotropy (whether electrons or ions) until pressure balance is achieved, the additional anisotropy of the pairs (electrons or ions) causes a pressure imbalanced depletion of the magnetic field which must be pressure compensated which happens mainly by sucking in additional cold non bouncing plasma of small magnetic moment from the surrounding. If this is forbidden, then one would expect that additional heating would be produced, and this would eat up the perpendicular anisotropy caused by the pairs, putting them back into the plasma and restoring pressure balance. In this case existence of pairs would be an intermediate state which leads to large mirror amplitudes (applicable to both electrons and ions).

4. Eq. 7, thanks this is trivial. Slightly changed.

5. Thanks for the language corrections.

---

## Editor Comment (EC2) · Nick Sergis (Editor) · 11 Sep 2019

The complete response to Reviewer 1 is posted under his/her review

---

## Editor Comment (EC3) · Nick Sergis (Editor) · 11 Sep 2019

The Topical Editor would like to thank all 4 reviewers for their constructive comments/suggestions and interesting questions to the authors, within the suggested time frame. While expecting the submission of a revised version of the manuscript, I would also like to thank the authors for their detailed responses.

---

## Author Comment (AC6) · 15 Sep 2019

We thank the reviewer for his comments. we are sorry for responding late, but a health problem intervened before we could reply.

We friendly ask R2 to consider the general comments which apply to all reviewers for the overlapping points.

Below are just a few concise remarks specific to R2.

1. why one needs to consider electrons? why not? first for curiosity which is a typical property of theoreticians or interested non-mechanical researchers. second because the growth rate of the general ion mirror mode also contains the anisotropy of electrons (e.g. noreen et al. 2018). so no need for any further justification.

[Figure]

just for a comment on shoji et al. by taking electrons as an isotropic fluid they eliminate all electron effects except quasineutrality. these simulations are nice but just show what has been put in (as all simulation do: they never discover anything beyond what is already contained in the equations they solve). with fluid electrons there will not be any electron effects nor ion sound waves for interaction. so electron pairing is excluded from the beginning.

we note in passing that our theory formally also applies to ions (which R2 is not aware of). we, however, consider only electrons, not ions which would require to check for waves they could interact with and for the different inertia. all going beyond the intention of our basic physics investigation as they would unreasonably increase the volume of the paper. the learned R2 might have believed we would not be aware of this possibility. we apologize of leaving the impression we would be ignorant.

we could add a number of other remarks on these simulations (which are a typical case of formally performing simulations, showing many figures but incompletely discuss the physics, clearly simulations always show something, i.e. solutions of the equations as figures. these need to be explained in terms of physics) but we refrain from it. we just say that in those sim's the large imm amplitudes are caused by the requirement of pressure balance which in the open system sucks in neutral plasma from the surrounding environment, without explanation why. in closed boundary sim's this will no be the case. R2's description is not further illuminating here. his last comment in his point 1 is superficial for the above reasons. no further comment on this.

2. this whole point is based on misunderstanding the physics. we have elucidated a physical effect here, not simply done some trivial simulation. R2 should read our general comments. true: we consider only parallel elstat waves as these apply to bouning particles in correct parallel physics (the only important physics her, which excludes any elmag waves). electromagnetic forces play no role in any generation of electric potentials, and there is no interest in radiation here; the comment on the mirror force is superficial, not withstanding that it is implicit to our main argument on bouncing particles. it is anyway surprising that it took half a century until in mirror modes the bounce is taken into account. We do not comment on langmuir and whistlers here as this is rhetoric nonsense notwithstanding that we commented on langmuir in the paper.

3. this is the reasonable point R2 makes of which we have been very well aware.

we refer R2 to the general comments. in brief, the gyrophase in elstat wave parallel resonance interaction clearly plays no role. there is no bunching of particles. in bounce any motion is gyroaveraged. so this point as made by R2 does not apply.

what applies but is not mentioned by R2 is our use of "coherent" which is meant to apply to the pairs locked near mirror points where their motion practically reduces to a common gyration (phase independent on this average bounce timescale as explained in the paper and general comments) at common speed $v_e$. this causes a surface or shell current which contributes a local diamagnetism, which however is very weak of the order of landau diamagnetism and negligible. we put this part in rewritten version into the appendix.

the essence of this and point 4 of R2 is correct: we have no true Meissner effect here, not only because the plasma is classical but because the diamagnetism is too weak (because ions are not fixed in a crystal here). thus susceptibility is simply pressure balance.

the important point is (still restricting for our purposes to electrons only, though it might also appy in similar vain to ions, as said above) that electron pairing in quasi-linear stability with vanishing ion anisotropy $A_i \approx 0$ and emerging large pair anisotropy $A_{e,pair} = T_e/m^* u^2 \gg 1$ gives rise to a substantial increase of the ion-mirror growth rate which without pairing woul not be not known.

hence in pairing, even for electrons only, the mirror mode continues growing fast beyond the quasilinear level even in closed boundary conditions.

it is clear that in simulations this requires inclusion of the fully kinetic electron dynamics. stabilisation under closed boundary conditions implies then destruction of the pair anisotropy which implies destruction of pairs and heating the electrons until new pressure balance at enhanced pressure, decreased internal magnetic field. otherwise it implies inflow of quasineutral plasma from the surrounding along the magnetic field to come up for the lack in pressure. this is the most probable mechanism. our expectation concerning susceptibility as interesting as it would have been, was overoptimistic. however the elucidation of this new and probably singular effect in plasma is a nice result.

once more thanks to R2 for the inquiry. though it was not his doubts but our own unrest with the original claim which forced us to rethink the physics of the reason for further growth with this satisfactory result.

we refrain from the demand of performing simulations which we leave to the efforts of simulationists. we dot see this as our duty which is not mechanically performing simulations but producing new possibly applicable physics. this has always been the duty of theorists until simulations came up which are now believed to be the non plus ultra but never produce any more insight than is contained in the input equations and boundary conditions and in addition requires the frequently lacking ingenuity of correct and exhausting interpretation of the figures obtained within the bounds of the input. just showing figures is not enough in physics.

---

## Author Response (AR1)

Thanks very much for your and the Reviewer's effort in the evaluation of this paper.

We upload the revised version of the MS. All substantial changes have been indicated in blue in the text. We have, however, not indicated minor changes in language and abbreviations suggested by the Reviewers and made by ourselves as these are of little importance.

The main points are clearly visible: they affect the pair anisotropy which contributes to substantially  modify the mirror growth. This has been discussed.

---

## Author Response (AR2)

The uploaded files differ from the last upload in a few minor points:

1. correction of a small number of typos
2. improvement of wording in a few cases and addition of commas for better structuring
3. introduction in Section 5 of 2 addtional subsection headings in the interest of clariy

I hope this is fine both with the editor and the journal. It does not introduce any change in the physics or argument.